# Stress, subjective wellbeing and self-knowledge in higher education teachers: A pilot study through bodyfulness approaches

Rosa-María Rodríguez-Jiménez[1,2]*, Manuel Carmona[3], Sonia García-Merino[4], Begoña Díaz-Rivas[5], Israel J. Thuissard-Vasallo[6]

1 STEAM School, Universidad Europea, Villaviciosa de Odón, Madrid, Spain, 2 European Association of Dance Movement Therapy, Berlin, Germany, 3 Institute for Regional Development (IDR), Universidad Castilla la Mancha, Albacete, Spain, 4 School of Exercise and Sports Sciences, Universidad Francisco de Vitoria, Pozuelo de Alarcón, Madrid, Spain, 5 El Viso Psicólogos, Madrid, Spain, 6 Faculty of Biomedical Sciences and Health, Universidad Europea, Villaviciosa de Odón, Madrid, Spain

* rosamaria.rojim@gmail.com

**Data Availability Statement:** All data are available from the Dryad Digital Repository: https://doi.org/10.5061/dryad.3j9kd51n1 Sharing link: https://

## Abstract

Work-related stress is a major cause of health problems worldwide. Faculty in educational institutions, including universities, also face high levels of stress, which undoubtedly affects their performance, level of personal satisfaction and wellbeing, and the relationship with students. Training interventions based on embodied learning can promote self-knowledge, emotional regulation and stress reduction, thereby increasing levels of psychological wellbeing. The present pilot study analyzed the impact of body awareness training in 31 university teachers using a controlled, randomized pre-post experimental design, with two experimental groups and a control group (n = 10). The two interventions were: Hatha Yoga (n = 11) and Dance Movement Therapy/Body Mind Centering (n = 10), which we have termed Body Movement Awareness. Variables related to body awareness, mindfulness, wellbeing, life satisfaction and stress were measured using self-perception tests. Cortisol levels, heart rate variability and sleep quality were also analyzed. Finally, participants' reflections were analyzed according to a qualitative approach. The results showed significant differences between the control group and the Hatha Yoga group in terms of stress reduction and wellbeing. The Body Movement Awareness group exhibited evidence of contributions on self-knowledge, communication and kinesthetic empathy, key elements in the educational field. Despite the inherent limitations of the study, the conclusions are encouraging and open new lines of research from embodied approaches that introduce creative movement and group experience as part of the process of emotional regulation and self-knowledge.

## Introduction

Leading organizations such as the International Labor Organization [1] and the European Agency for Safety and Health at Work [2] recognize that work-related stress is the epidemic of the 21st century and a major cause of health problems in Europe. Stress and, more specifically,

datadryad.org/stash/share/3YTJpA-RFXa4AU3N31jDKhMFs_MWeaV2FRK5tBCX-Pc.

**Funding:** Universidad Europea de Madrid funded this research, under grant number UEM27. https://universidadeuropea.com/ Dr. Manuel Carmona enjoyed a Ramón y Cajal Fellowship from the Spanish Ministry of Science, Innovation and Universities for the (RyC-2014-16307). https://www.ciencia.gob.es/ The funders had no role in study design, data collection and analysis, decision to publish, or preparation of the manuscript.

**Competing interests:** The authors have declared that no competing interests exist.

work-related stress, is part of everyday life in our society. Accordingly, understanding the relationship between stress and health is essential to implement stress reduction programs that will aid in fostering healthier work environments.

Stress results from the interaction between the individual and their environment, which is triggered when faced with a situation that is perceived as threatening and for which sufficient resources may not be available [3]. The intensity of the response involves both physiological (endocrine) processes and emotional/psychological processes [4–6]. Highly demanding work environments require the worker to maintain an almost uninterrupted state of alertness and activation over prolonged periods. In this situation, the health of the worker, as well as that of the organization, is at risk of being impaired [2, 7]. Continuous and high-level stress can lead to "burnout syndrome", defined as a state of mental, physical and emotional exhaustion produced by involvement in work in emotionally-demanding situations over long periods [8].

According to Einsiedel & Tully [9], five categories of symptoms that negatively affect health can be distinguished in burnout syndrome: physical symptoms, excessive behavior, emotional problems, and relevant changes in personal relationships, and in attitudes, values and beliefs. According to the transactional model of Lazarus [3, 10], not everyone responds equally to the different demands of the environment, which modulates the occurrence of stress-related symptoms, and also explains the concept of subjective wellbeing [11]. From the perspective of Diener [12], subjective wellbeing is defined as the affective balance and perceived vital satisfaction––that is, the balance of emotions plus the global judgment on one's own life. This concept has attracted numerous studies and multiple scales and questionnaires for its measurement [13–15].

With regards to work-related stress, an individual's perception of wellbeing is implicitly linked both to the personal response to stress and to the meaning that they give to health and wellbeing. In this context, positive emotional management will contribute to a greater perception of personal wellbeing and, consequently, to a greater protection against stressful events and a better state of health [16]. Indeed, many of the studies examining these issues report a correlation between positive emotions and cognition and physical, mental and social health [17, 18].

Work-related stress in the educational field is considered highly relevant [7]. The last decade has witnessed the increasing focus on teacher stress, including at the university level [19–21]. Analyses of several studies on work-related stress in this profession in Spain reveal that many Spanish university teachers present with symptoms of work-related stress and average burnout rates, which negatively affect their job satisfaction, productivity, and physical and emotional health [22, 23].

Given the above, it is important to better understand the configuration and complexity of the teaching profession today [24], in particular at the university level [25]. University teaching is composed of professional competencies that include not only technical knowledge of each subject of study but also communication skills, empathy, didactical resources, emotional management and development of interpersonal relationships, among others. This definition of the profile of university teachers is widely reinforced by the Bologna reform, which culminated in 2020, with a paradigm shift in the teaching-learning system [26]. It is, however, difficult to train in competencies if no previous training along such lines has been undertaken. This situation might lead university teachers to feel outperformed in terms of knowledge, skills or requirements, becoming psychosocial risk factors for increased levels of work-related stress [27].

Some successful experiences have been put into practice for interventions on stress in higher education. Most of them originate from approaches based on the Mindfulness-Based Stress Reduction program, promoted by Jon Kabat-Zinn [28]. Programs with these

characteristics are taught at universities such as Georgetown, Harvard and Stanford, and studies with teachers of different levels and educational fields as well as with students have shown encouraging results [29–31]. Other proposals include mind-body therapies such as Yoga or TaiChi [32, 33].

Some essential aspects of these types of intervention include training in a better knowledge of oneself and full attention in the here and now (through meditation), and giving considerable relevance to the concept of body consciousness or awareness. Within this framework, body consciousness is understood as the identification and description of physical sensations and perceptions that bring information to the cognitive-emotional sphere [34, 35]. An intervention based on this approach allows the individual to explore the dimensions of their own corporeality. As indicated by Mehling [34], body consciousness is an inseparable aspect of the consciousness of an embodied self that is constructed in interaction with the environment. It is the product of a dynamic process that helps to build oneself, including cognitive assessment and attitudes, beliefs, experiences and learnings in the personal and relational context [35, 36].

It seems reasonable that a body consciousness approach could be applied to work-related stress in the university teaching environment, which is determined not only by the highly emotional demand of the environment, but also by the interpersonal relations with students, a key part of this environment. Generally, it is in the awareness of experience, in transaction and action, where potential conflicts arise; therefore, it is not only a question of improving the stress symptoms of university teachers––for which meditation or Yoga might contribute––but also of providing them with more and better competence tools to interact with their work environment, focusing also on interpersonal relations [37, 38] through increasing their body consciousness. This is what an enactive and bodyfulness approach will provide in the study of embodiment: the body is not something considered as something "to possess", but as something "that is" and that develops in continuous relationship with the environment [39, 40].

In addition to approaches such as Yoga or mindfulness, artistic approaches offer opportunities for both stress regulation and the development of self-awareness and self-knowledge in a relational context [41, 42]. Indeed, the use of creative tools has proven effective in clinical and non-clinical contexts for reducing stress and anxiety and promoting higher levels of wellbeing and health [43–48]. In this context, dance movement therapy (DMT) and body and mind centering could be useful tools to help modulate stress levels in teachers, as demonstrated previously [43, 49]. DMT belongs to the so-called creative therapies and uses the relational and therapeutic use of dance and movement to further the physical, emotional, cognitive, social, and cultural functioning of a person [50]. It is based on embodiment theories, body awareness and experiential meanings of gesture, posture and movement, and combines psychoanalytical knowledge of the body's memory, creative processes, analysis and observation systems with neuroscientific contributions on the biunivocal relationship between the motor and the cognitive-emotional systems [51–53]. DMT has traditionally been used in the clinical setting, but is now increasingly incorporated into prevention and health promotion programs for non-clinical populations [43, 54].

Body-Mind Centering® (BMC) [55] is a discipline in the field of somatic movement education and therapy. It is an experiential approach based on the embodiment and application of anatomical, physiological, psychophysical and developmental principles, utilizing movement, touch, voice and mind. Its uniqueness lies in the specificity with which each of the body systems can be personally embodied and integrated, the fundamental groundwork of developmental re-patterning, and the utilization of a body-based language to describe movement and body-mind relationships. The study of BMC is a creative process in which embodiment of the material is explored in the context of self-discovery and openness.

Providing the teacher with ongoing training that integrates these concepts can foster meaningful and transformative learning [56] that can contribute to the teacher's own wellbeing and also improve communication and relationships with students [57, 58]. This requires training interventions adapted to the academic institutions, the timeframe, and the needs of the teachers. The aim of this pilot study was to analyze the impact of a training intervention through two programs that may be complementary in the development of body awareness but have their own characteristics: Hatha Yoga (HY) and Body Movement Awareness (BMA). BMA will be the term used for the program developed with DMT and BMC. The study highlights differences between the two approaches and underlines the contributions that embodied approaches have on teachers' well-being. The general objective was divided into two specific objectives. Firstly, to assess the impact of two different intervention programmes through different quantitative measures related to well-being, life satisfaction and stress. Secondly, to explore the differences between the two approaches in relation to self-awareness, self-knowledge and relational aspects through qualitative data. The qualitative data shed light on each participant's experience and their perception of the contribution of the different approaches to their well-being. The development of self-awareness and its relationship to health are dynamic processes incorporating a variety of personal and relational dimensions that may be difficult to represent by quantitative measures alone. The relational component, not included in others embodied approaches, is a key aspect in education [22–23]. The novelty of this paper lies in its introduction in higher education that until now is scarce. For all the above reasons, mixed approaches can enrich and increase knowledge in this area [34–36].

## Materials and methods

### Sample and setting

Participants were recruited from the teaching staff of the Universidad Europea de Madrid (UEM), Madrid, Spain, through an on-line invitation sent by the Health Prevention Department of the University. Participation was voluntary, and there was no academic reimbursement. The inclusion criterion consisted of being a university professor with at least two years of experience in the institution and not having completed a program of the same characteristics in the last year. As an exclusion criterion, we used the Physical Activity Readiness Questionnaire (PARQ+), a 7-step screening questionnaire to determine whether potential participants could undertake the programs designed safely. A second exclusion criterion was to have already carried out a similar activity to those proposed for at least six months on a weekly basis. Fig 1 shows the the phases of the randomised trial of three groups (that is, enrolment, intervention allocation, follow-up, and data analysis) according CONSORT guidelines.

The pilot study was approved by the Ethics Committee of Medicine Research of the Community of Madrid (CEIM-R; EC13.17) and by the Ethics Committee of the Universidad Europea de Madrid (UEM2017-27). Selected participants signed an informed consent and a code was assigned to each to maintain confidentiality. Participants were randomly assigned to a control group (group 1, n = 10) and to one of two experimental groups (groups 2 (n = 11) and 3 (n = 10)). Groups 2 and 3 were informed by e-mail of the date for practical sessions of HY or BMA, respectively. They were also informed that the sessions would be held during working hours and within the university facilities. Human resources department validated the sessions as part of the compulsory training program that all teachers must do annually.

### Procedure

We applied a mixed experimental methodology to gather data [59]. Mixed-method approach offers the best opportunity for addressing research questions by integrating both quantitative

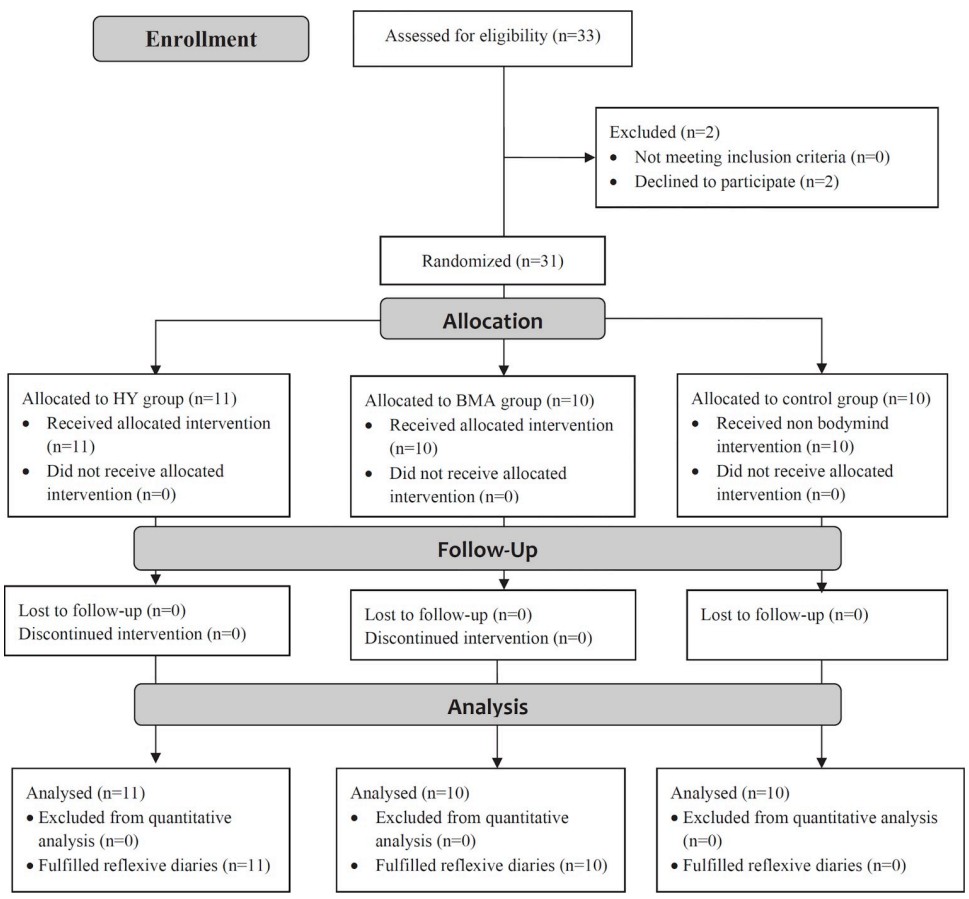

**Fig 1. Consort flow chart of control and intervention groups.**

and qualitative methods. Additionally, qualitative analysis allows for a better understanding of the process. Including thoughts, feelings, sensations and perceptions of participants during the sessions give us a deeper knowledge of what happened and allows us to identify areas for improvement. Physiological variables representative of the stress level of the participants: cortisol levels in saliva, heart rate variability and sleep quality, were recorded in all three groups. Levels of activity were measured before the intervention to detect differences among the groups. Levels of body awareness, mindfulness, subjective wellbeing, life satisfaction and perceived stress were measured using five validated psychometric tests according to a pre-post type methodology. The complete program was developed in the following three phases:

a) Phase pre-test: Once the participants have passed the exclusion criteria, they fulfilled the six questionnaires, saliva samples were collected to establish basal cortisol levels, and measures of heart rate variability (HRV) and sleep quality were performed.

b) Intervention: The intervention program was performed over two months. Control group 1 did not perform any activity and groups 2 and 3 performed the HY and BMA programs, respectively (1 weekly session of 90 minutes duration, 8 sessions in total). Both programs shared some common work dynamics at the beginning and at the end of each session. The sessions took place from 14.00 h to 15.30 h Central European Time in a Pilates room (45 m$^2$) for the HY program and in a Tatami room (60 m$^2$) for the BMA program, both located at the Sports Building in the Universidad Europea de Madrid. The specified rooms guaranteed a warm and quiet atmosphere and confidentiality, and were equipped with music, mats and

**Table 1. Structure and objectives of the Hatha Yoga and Body Movement Awareness programs.**

| Program | Hatha Yoga | Body Movement Awareness |
|---|---|---|
| **Check-in (1)** | Introduction to body listening. | |
| **Warm-up (2)** | Joint mobilization. | Confidence in space, own body and group. |
| **Development (3)** | Connection with the center. Alignment and stability. Spine mobility Internal organs activation. | Body structure, limits and muscular tension. Movement qualities and patterns of one's own or others. Imitation, body listening & kinesthetic empathy. Integration of sensation, emotion and cognition. |
| **Relaxation (4)** | Returning to the calm. | |
| **Closure (5)** | Personal reflection (diary) and group discussion. | |

Note: Sections 1, 4 and 5 were common to both programs.

different type of materials such as elastics, different forms of fabrics, soft balls and balloons. Both programs were taught by specialists in Yoga and Dance Movement Therapy respectively, both with more than 20 years of experience in the field. Reflective diaries were collected weekly for both experimental groups, and saliva samples were collected at the end of the sessions.

c) Phase post-test: After the intervention phase, the three groups were again subjected to the measurement of the same physiological variables and to the five questionnaires. Saliva samples were collected again from the three groups one week after the end of the intervention and 6 weeks later.

**Hatha Yoga and Body Movement Awareness intervention.** The interventions followed the structure shown in Table 1.

The HY program consisted of the practice of postures and counter postures aimed at increasing flexibility and strength, and with a focus on body alignment and lateralization. The sequence of the exercises was practically identical across all sessions with slight adaptations according to the level of progress and needs of the participants (Table 2). The work was done statically on a Yoga mat and there was no interrelation between participants.

The BMA program consisted of a sequence of different sessions with the exception of the common sessions at the beginning and at the end. Each session revolved around a theme using different moving resources to work on the acquisition of body consciousness (Table 3). Most of the work was done in movement across the space, with moments of interaction with others (dyads or the whole group).

**Table 2. Hatha Yoga program, asanas and exercises performed.**

| OBJECTIVES | ASANAS AND EXERCISES |
|---|---|
| **Introduction to body listening Attention to pelvis-thorax-head axis** | Standing position and breathing exercises: Tadasana. |
| **Joint mobilization** | Urdhva Hastasana / Tadasana variations. |
| **Connection with the center, alignment and stability** | Standing postures: Utkatasana / Uttanasana / Anjaneyasanan (knee up) / Virabhadrasana II / Utthita Parsvakonasana / Utthita Trikonasana / Uttanasana. Center postures with neutral pelvis: Adho Mukha Svanasana / Adho Mukha Virasana. |
| **Spine mobility and internal organs activation** | Flexion and torsion postures: Janu Sirsasana / Baddha Konosana / Paschimottanasana / Ekapada-Pavanamuktasana / Pavanamuktasana / Jathara Parivartanasana. |
| **Returning to the calm** | Rotation of the consciousness through the whole body: Savasana. |

**Table 3. Body movement awareness program, description of the themes followed and the dynamics used.**

| SESSION / THEME | DYNAMICS | |
| --- | --- | --- |
| | **Warming up** | **Development** |
| 1 / Knowing the group | Breathing exercises / Body scan / Joint movement / Sensory dynamics | Games of knowledge in the group / Construction of figures in a collaborative way. |
| 2 / Space and time (Laban Analysis) | Vertical body scan / Breathing / Joint movement / Stretching of muscle chains | Dynamics of understanding the qualities of movement in space (focus) and use of time (decision making). |
| 3 / Motor actions (Laban Analysis) | Body scan /Observation of weight and supports / Shared connection | Working in pairs on the actions of fighting-indulgence /Realization of free movement according to the above actions. |
| 4 / Anatomical structure: (skin, muscles, bones) | Body scan / Breathing /Activation and gentle joint mobilization | Recognition of different anatomical structures through physical contact, in pairs / Free movement based on the differentiating qualities of each structure. |
| 5 / Grounding & support | Contact of feet on the floor / Support in pairs | Games of endurance, letting go and supporting weight. Differences between actions of fight and indulgence in movement. |
| 6 / Muscle chain | Exercises of self-contact | Recognition and observation of types of muscle chains. |
| 7 /Body boundaries / Needs | Square breathing / Visualization / Individual mobilizations with limitations | Play in pairs with limits in degrees of freedom. Use of the concept of limit in the movement in space. |
| 8 / Kinesthetic empathy | Free Mobilization / Use of rhythm observation and listening to the other | Dynamics of attunement and kinesthetic empathy (in pairs and group). |

## Outcome measures

The different quantitative and qualitative variables and instruments used are listed below. Firstly, patterns of physical activity (GPAQ) and suitability for physical activity (PARQ+), the latter being used as an exclusion criterion. Secondly, the variables collected through psychometric tests for the pre-post study: levels of body awareness (BAQ), mindfulness (FFMQ), subjective wellbeing (WHO-5), life satisfaction (SWSL) and perceived stress level (PSS) are presented. Then, the physiological variables such as the heart rate variability, the sleep quality and cortisol levels, are shown. Finally, the qualitative measures, that is, the participants' perceptions.

**Global Physical Activity Questionnaire.** GPAQ [60] is a validated questionnaire of 16 items assessing physical activity patterns in the domains of work (minutes working, studying or doing housework, transport (minutes of walking and bicycling), leisure-time (minutes doing sport, fitness activities or other activities in leisure-time), and sedentary behavior (minutes spent sitting or lying) in a typical week. A Spanish version (GPAQ v2) was used in an interview-administrated mode. The questionnaire was development by the World Health Organization (WHO) in 2002 [61] and, following the criteria of this organization, participants are considered "insufficiently active" if they did not complete at least 150 minutes of moderate-intensity or 75 minutes of vigorous intensity or an equivalent combination of moderate and vigorous intensity physical activity weekly, achieving at least 660 Metabolic Equivalent Task (MET)-minutes.

**Physical Activity Readiness Questionnaire.** The PARQ+ [62] comprises a range of questions to identify any possible restrictions or limitations on physical activity participation. The procedure is detailed as follows: the participant answers the 7 questions in part one. If the answer is no to all of the questions, physical activity participation is cleared unrestricted.

However, if the participant answers yes to 1 or more questions in part one, they continue with the second part of the questionnaire (which contains questions on specific chronic medical conditions). If the participant answers yes to 1 or more questions in the second part, they are referred to a qualified exercise professional to identify possible limitations on physical activity participation. The PARQ+ screening is valid for 12 months and takes approximately 5 minutes to complete.

**Body Awareness Questionnaire (BAQ).** The BAQ [63] measures attentiveness to normal, non-emotive internal bodily processes and sensations, particularly, sensitivity to bodily cycles and rhythms, small changes in normal functioning, and anticipation of bodily reactions. The instrument contains 18 items scored on a 7-point Likert scale ranging from 1 (not at all true about me) to 7 (very true about me). The original version of the BAQ has four scales: note responses or changes in body process; predict bodily reaction; sleep-wake cycle; and onset of illness; but normally a total score is calculated. A Spanish version was used and the procedure for its final drafting is described in detail in S1 Appendix. The questionnaire is appropriate for both men (alpha coefficient = 0.82) and women (alpha coefficient = 0.80), has good test-retest reliability (r = 0.80), and has discriminant validity and stability in factor structure (Shields et al., 1989).

**Five Facet Mindfulness Questionnaire (FFMQ).** This instrument [64] contains 39 items scored on a 5-point Likert scale ranging from 1 (not at all true about me) to 5 (very true about me). It is based on a factor analytic study of five independently developed mindfulness questionnaires. The analysis yields five facets that appear to represent elements of mindfulness as it is currently conceptualized: observing, describing, acting with awareness, non-judging of inner experience, and non-reactivity to inner experience.

*WHO (five) Wellbeing Index (WHO-5).* WHO-5 [65] is a short self-reported measure of current mental wellbeing that consists of five statements, which respondents' rate according to a scale (in relation to the past two weeks). It is scored on a 5-point Likert scale ranging from 0 (at no time) to 5 (all the time). The WHO-5 has been found to have adequate validity in screening for depression and in measuring outcomes in clinical trials. The measure has good construct validity as a unidimensional scale measuring wellbeing in younger and elderly populations [65]. The total raw score, ranging from 0 to 25, is multiplied by 4 to give the final score, with 0 representing the worst imaginable wellbeing and 100 representing the best imaginable wellbeing.

**The Satisfaction with Life Scale (SWLS).** SWLS [66] is a short 5-item instrument designed to measure global cognitive judgments of satisfaction with one's life, and is scored on a 7-point Likert style response scale ranging from 1 (strongly disagree) to 7 (strongly agree). The possible range of scores is 5–35, with a score of 20 representing a neutral point on the scale. Scores of 5–9 indicate that the respondent is extremely dissatisfied with life, whereas scores of 31–35 indicate that the respondent is extremely satisfied. The alpha coefficient for the scale ranges from 0.79 to 0.89, indicating that the scale has high internal consistency. The scale also has good test-re-test correlations (0.80 over a month interval).

**Perceived Stress Scale (PSS).** The PSS [13] is the most widely used measure of global perceived stress and is a robust predictor of health and disease. We used a Spanish version [67]. The 10-item PSS measures global perceived stress experienced across the past 30 days on a 5-point scale ranging from 0 (never) to 4 (very often). Total scores range from 0 to 40. Internal consistency reliability was $\alpha = 0.84$ for Spanish responders.

**Heart rate variability.** To determine the values of HRV, the R-R intervals between R waves (peak) of successive heartbeats were measured using the heart rate monitor Polar V800 (Polar, Finland). Data were analyzed using the software Kubios (Kuopio, Finland) to determine the predominance of the sympathetic or parasympathetic nervous system. The

measurements were performed in the laboratory in the early hours of the morning with the following protocol: the individual remained lying down for 5 minutes, the measures were then collected over 10 minutes in the same position, before beginning the program and at the end.

**Quality, efficiency and total sleep time.** Variables were measured using the Actigraph GT9X Link Monitor. The accelerometer generates resting-activity patterns and thus allows estimation of sleep-wake cycles. Actilife 6F software was used for the analysis of the information relative to the period asleep, total time of sleep and the moment in which participants fall asleep. Participants placed the device in their non-dominant hand for 3 consecutive days. The information was obtained just before starting the full program and at its end.

**Saliva cortisol levels.** On the first day of sampling, the participants were instructed how to correctly take the sample at the same time of the day (detailed in S2 Appendix). When the participants arrived at the sessions in which saliva is sampled, they were provided with a 15-ml sterile tube and an indelible marker so that they could write their identification code and the date of the day on the outside of the tube. The participants deposited a saliva sample of at least 1 ml volume and gave it to the responsible person of the research team. Saliva samples were frozen at -20˚C until cortisol levels were determined in duplicate using an ELISA test (DeMe-Tec diagnostics, Kiel, Germany) on the EZread 400 microplate reader spectrophotometer (Biochrom Ltd., Cambridge, UK).

**Participants' perceptions.** Participants in the intervention groups (n = 21) completed reflective diaries [68] after each practice session as shown in Fig 1. The diaries had a personal and non-directed nature, and participants were invited to freely express aspects related to their thoughts, sensations, perceptions and feelings they experienced during the sessions. Diaries helped them to be aware of their learning process through the opportunity to express it, providing them a greater involvement in the program.

## Data analysis

Continuous variables are presented as mean and standard deviations or median and interquartile range according to the parametric characteristics of the variables. For qualitative variables we used absolute (n) and relative (%) frequencies. After confirming the parametric distribution of the quantitative variables with the Shapiro-Wilk test, an ANOVA test was used to verify if there were differences among the groups before the intervention. Descriptive and paired-sample t-tests were conducted to compare differences pre- to post-intervention in each group and Cohen's d effect sizes were calculated. Values of 0.2, 0.5, and 0.8 were interpreted as small, medium, and large effect sizes, respectively (Cohen, 1988) [69]. For those variables where the test of normality was accepted, the effect of the intervention was analyzed using parametric ANOVA for repeated measures. The significance level for all tests was $P<0.05$. The Statistical Package for the Social Sciences (SPSS, IBM, Armonk, NY), version 25.0 for Windows, was used for statistical analysis.

The reflective diaries were transcribed and systematically read and were labeled with the participant code and chronologically numbered by session. The diaries were reread until saturation to ensure that all aspects were covered in the categories. Research triangulation was applied, allowing a researcher outside the interventions to independently perform the qualitative analysis according a deductive paradigm [70]. It was based on a conceptual matrix of body awareness proposed by Mehling [71] of the following dimensions: a) perceived body sensations; b) attention quality; c) attitude and d) mind-body integration. These dimensions were taken as initial categories within the study. Subsequently and after successive meetings, the final structure of metacategories and categories was decided. The qualitative analysis was done using the NVivo software (version 12).

## Results

### Characterization of participants

The social characteristics of the participants are shown in Table 4. The sample included 31 participants (11 men, 20 women), with more than 75% of the participants married and 65% having children. Participants had a wide range of experience in terms of years (mean 10 years). Slightly more than half of participants (58%) had part-time contracts, and some (29%) had another employment engagement. Examination of the number of work hours in the control and intervention groups showed that 80% of the former had a partial contract compared with 58% of the latter. By contrast, ~50% of all participants in the two intervention groups had a partial contract.

None of the potential participants who responded to the PARQ+ were excluded from the study. The screening tool requires that the participant answers YES to at least one of the seven initial questions of the test to continue with further questions and suggest a medical check-up before doing any kind of physical activity. This was not the case as all participants answered NO to all 7 initial questions. Regarding physical activity assessed with the GPAQ, there were no significant differences between the randomized groups, although the control group showed the highest MET value with 1220 min/week and the BMA group had the lowest MET value with 720 min/week (Fig 2).

The variables pre-intervention were tested by means of an ANOVA test (Table 5). No significance differences among groups were observed for any of all parameters.

**Table 4. Social characteristics of the participants.**

| Items | Total (31) | Control (10) | HY (11) | BMA (10) |
|---|---|---|---|---|
| **Sex**[1] | | | | |
| Men, n (%) | 11 (35.5) | 4 (40.0) | 3 (27.3) | 4 (40,0) |
| Female, n (%) | 20 (64.5) | 6 (60.0) | 8(72.7) | 6 (60.0) |
| **Age**[2] | 43.3 ± 7.5 | 43.5 ± 6.5 | 45.0 ± 6.9 | 41.1 ± 9.2 |
| **Civil status**[1] | | | | |
| Single | 6 (19.4) | 1 (10.0) | 2 (18.2) | 3 (30.0) |
| Married | 24 (77.4) | 9 (90.0) | 9 (81.8) | 6 (60.0) |
| Divorced | 1 (3.2) | 0 (0.0) | 0 (0.0) | 1 (10.0) |
| **Children**[1] | | | | |
| No | 11 (35.5) | 3 (30.0) | 4 (36.4) | 4 (40.0) |
| Yes | 20 (64.5) | 7 (70.0) | 7 (63.6) | 6 (60.0) |
| **Number of children**[3] | 1.0 [2.0] | 1 [2.0] | 2 [2.0] | 1 [1.3] |
| **Contract type**[1] | | | | |
| Partial | 18 (58.1) | 8 (80.0) | 5 (45.5) | 5 (50.0) |
| Full-time | 13 (41.9) | 2 (20.0) | 6 (54.5) | 5 (50.0) |
| **Years of experience** | 10.2 ± 7.0 | 9.7 ± 8.3 | 11.8 ± 6.8 | 8.9 ± 6.0 |
| **Current job state**[1] | | | | |
| Associate Professor | 15 (48.4) | 6 (60.0) | 6 (54.5) | 3 (30.0) |
| Full Professor | 15 (51.6) | 4 (40.0) | 5 (45.5) | 7 (70.0) |
| **Other employment**[1] | | | | |
| No | 22 (71.0) | 5 (50.0) | 8 (72.7) | 9 (90.0) |
| Yes | 9 (29.0) | 5 (50.0) | 3 (27.3) | 1 (10.0) |

[1] Mean and percentages (in parenthesis) are shown for most of the items

[2] Mean and standard deviation (in parenthesis) are shown for years of experience

[3] Medium and interquartile (in brackets) are shown for number of children.

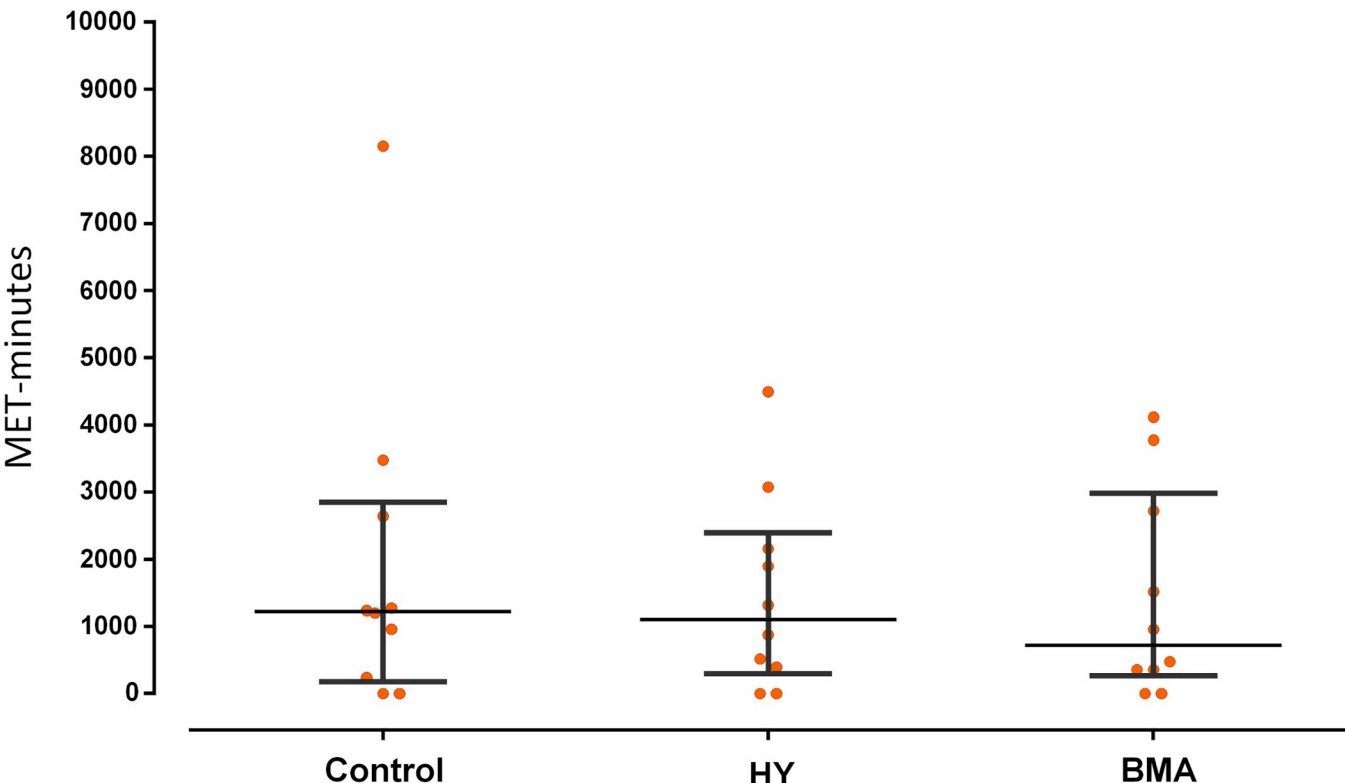

**Fig 2. Mean and standard deviation for GPAQ pretest.** HY, Hatha Yoga; BMA, Body Movement Awareness; MET min, minutes of Metabolic Equivalent Task per week.

### Psychometric test results

Evaluation of the overall (pre-post test) results provided by the BAQ test revealed significant differences only in the HY group, with a high effect size (Cohen's d = 4.063) (Fig 3). A trend was observed in the BMA group with a medium effect size (Cohen's d = 0.636), but it was no significant.

**Table 5. Mean, standard deviation and results of ANOVA test for the three groups (Control, HY and BMA) and all the variables before intervention.**

| Variables | Control (n = 10) | HY (n = 11) | BMA (n = 10) | p-value* |
|---|---|---|---|---|
| Body Awareness (BAQ) | 81.14 ± 19.68 | 76.00 ± 19.60 | 75.36 ± 22.17 | 0.831 |
| Mindfulness (FFMQ) | 128.56 ± 19.26 | 123.80 ± 17.91 | 128.82 ± 18.48 | 0.793 |
| Subjective wellbeing (WHO-5) | 58.67 ± 18.76 | 53.60 ± 17.71 | 58.91 ± 18.43 | 0.766 |
| Life satisfaction (SWLS) | 23.56 ± 3.81 | 25.70 ± 4.35 | 26.00 ± 3.97 | 0.371 |
| Perceived stress (PSS) | 20.13 ± 7.51 | 18.40 ± 6.43 | 13.09 ± 5.38 | 0.054 |
| Sleep efficiency (%) | 84.98 ± 5.20 | 84.18 ± 7.51 | 87.95 ± 4.79 | 0.339 |
| Low frequency (LF, nu)** | 54.95 ± 14.45 | 52.90 ± 15.00 | 46.17 ± 19.47 | 0.456 |
| High frequency (HF, nu)** | 44.94 ± 14.38 | 47.07 ± 14.99 | 53.72 ± 19.51 | 0.457 |
| Ratio LF/HF (ms$^2$) | 1.55 ± 1.23 | 1.55 ± 1.06 | 1.14 ± 0.92 | 0.724 |
| Cortisol value | 2.97 ± 0.80 | 2.97 ± 1.07 | 3.38 ± 1.46 | 0.623 |

* Significative differences (p<0.05) among the three groups

**Low frequency (LF) and high frequency (HF) are normalized components of heart rate variability

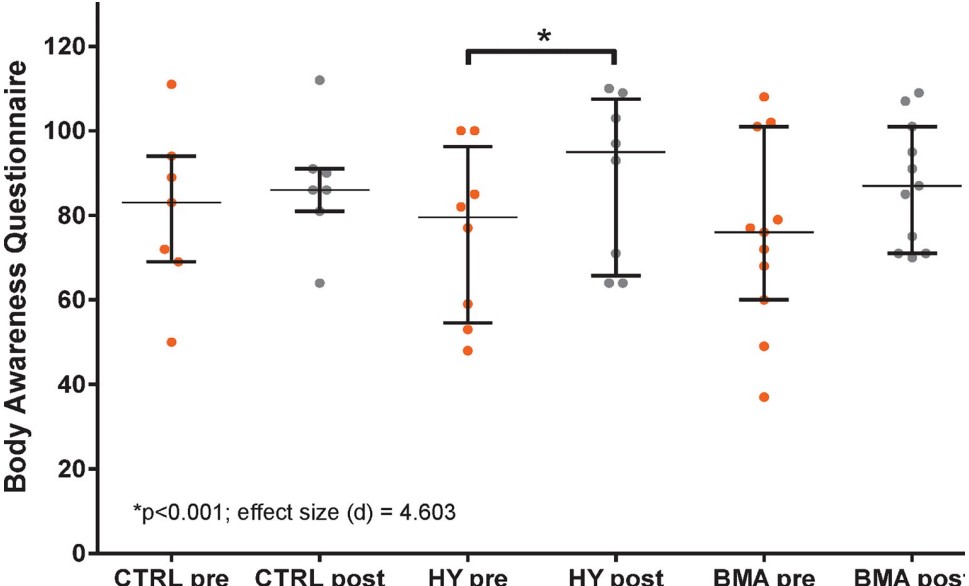

**Fig 3. Mean and standard deviation for BAQ pre-post test.** HY, Hatha Yoga; BMA, Body Movement Awareness; *Statistically significant; d-value represents Cohen's effect size.

To clarify where these differences lay, we analyzed the 4 scales that subdivide the test. Results for the four scales showed significant differences for HY group. The scores in the HY group for the four scales increased significantly after the intervention. The same trends were observed in the BMA group but only the scale body reactions was significant.

There were an increase post intervention in mindfulness, life satisfaction and wellbeing for the two groups but only changes in the subjective wellbeing were significant for the HY group (p = 0.008) and a trend in life satisfaction was observed in BMA group with a high size effect (p = 0.051; d = 0.764). The HY group showed significant reduction in perceived stress (p = 0.012) after the intervention (Fig 4). Additional details about the results of the ANOVA test post intervention and paired-sample T-test pre and post intervention for all the variables and groups are presented in S3 Appendix.

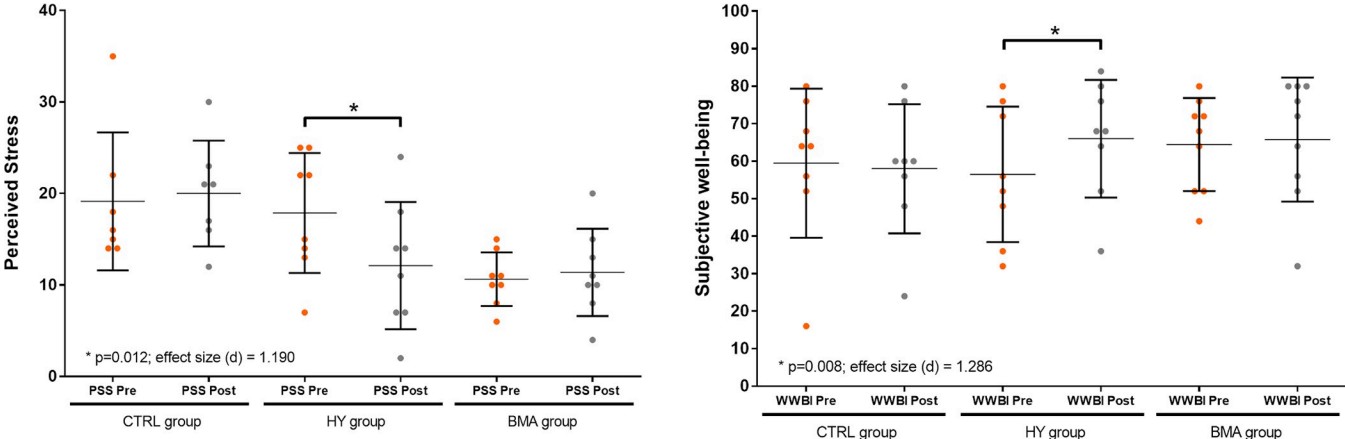

**Fig 4. Mean and standard deviation for WHO-5 and PSS pre-post results.** HY, Hatha Yoga; BMA, Body Movement Awareness; *Statistically significant; d-value represents Cohen's effect size.

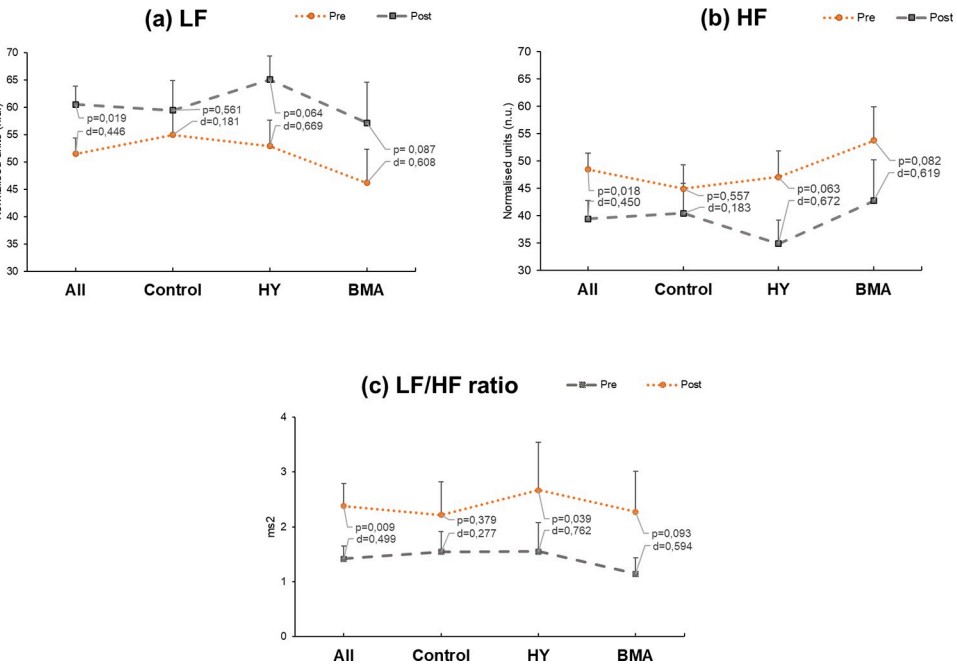

**Fig 5.** Pre-post comparison within the frequency domain of the following heart variability parameters: a) Low Frequency (LF), b) High Frequency (HF), and c) their ratio LF/HF. Mean and standard deviation are shown. HY, Hatha Yoga; BMA, Body Movement Awareness; Statistically significant at p<0.05; d-value represents Cohen's effect size.

## Heart rate variability

With regards to HRV, we first examined the frequency domain by analyzing the Low Frequency (LF) component, which correlates with the activity of the sympathetic and parasympathetic nervous system, and the High Frequency (HF) component, which correlates with parasympathetic activity. The ratio LF/HF is the most suitable parameter for estimating sympathetic-vagal equilibrium because of the controversy that exists in the interpretation of the LF and HF parameters in isolation, especially the former [72].

Analysis of the whole sample revealed that the LF parameter increased significantly in pre-post test analysis irrespective of the group (p = 0.019; Fig 5). This trend remained when we examined individual groups, although in the case of the control group there was no significant change (p = 0.666), and there was only a weak tendency in the other two groups (HY, p = 0.064; BMA, p = 0.087), both with a medium effect size (Cohen's d >0.6) (Fig 5).

A similar relationship was observed for the HF parameter, which decreased significantly in the whole sample (p = 0.018) but did not reach significance in the intervention groups (HY, p = 0.063; BMA, p = 0.082) (Fig 5). Both interventions reached a medium effect size (Cohen's d = 0.672 [HY]; d = 0.619[BMA]).

As expected, the LF/HF ratio increased in pre-post test analysis in all cases (Fig 5). This increase was again significant for the whole sample (p = 0.009, d = 0.499) but not in the control group (p = 0.451, d = 0.236). It was also significant for the HY group (p = 0.039, d = 0.762). There was a trend for significance in the BMA group (p = 0.093, d = 0.594). Either program had an impact with respect to the control in the LF/HF ratio.

With regards to the time domain parameters for HRV, no significant differences or noticeable effect sizes were reached for the standard deviation of normal-to-normal intervals (SDNN) or the root mean squared of successive differences (RMSSD) (not shown).

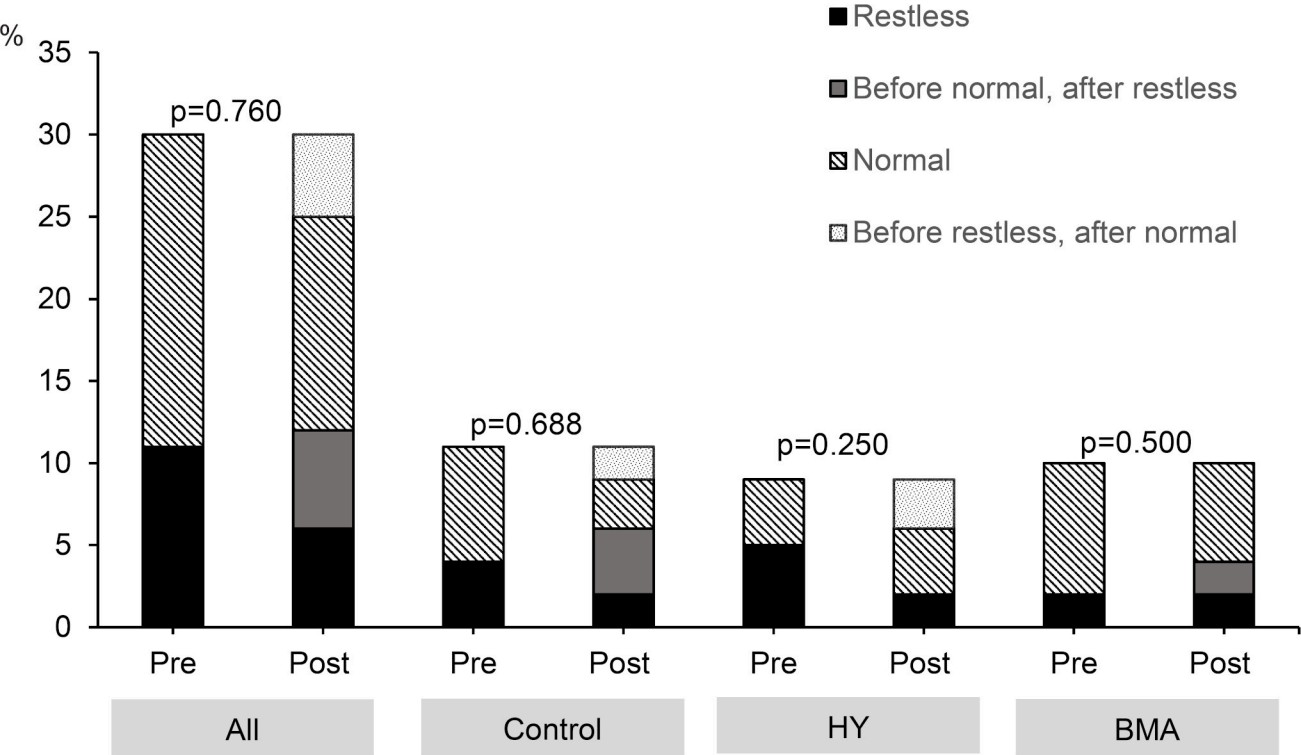

**Fig 6. Sleep analysis results.** HY, Hatha Yoga; BMA, Body Movement Awareness.

## Quality and efficiency of sleep time

No conclusive results were obtained for sleep efficiency (Fig 6). We analyzed rest-activity patterns, considering people who did not rest well as those who rested <85% of the total time of sleep. No differences were found when considering the whole group or when participants were stratified by intervention. Fig 6 shows a categorized analysis of how many participants changed their classification (normal pattern >85%; restless pattern <85%) after the intervention. In the control group, four participants who were in the normal pattern switched to the restless pattern and two others switched in the opposite direction. In the HY group, three participants who did not rest well changed to a normal pattern after the intervention; in the BMA group, two participants who showed a normal pattern before the intervention changed to a restless pattern.

## Cortisol levels

Analysis of saliva cortisol at baseline revealed similar mean levels in the control (2.97 ± 0.78 ng/ml) and HY (2.97 ± 1.07 ng/ml) groups, and a slightly higher mean level in the BMA group (3.39 ± 1.46 ng/ml), which also showed a larger fluctuation of values around the mean (Fig 7). No significant differences were found between the baseline values by one-factor ANOVA analysis.

Mean cortisol levels in the HY and BMA groups decreased at the end of the intervention period and showed less variability (Fig 7B and 7C). By contrast, cortisol levels in the control group increased (Fig 7B and 7C). None of these changes were, however, significant.

The follow-up of the intervention groups was extended one month after the end of the program. While no significant differences were found in cortisol levels (Fig 7B and 7C), we found a tendency for increased levels in both groups. Of note, the increase in cortisol was smaller in

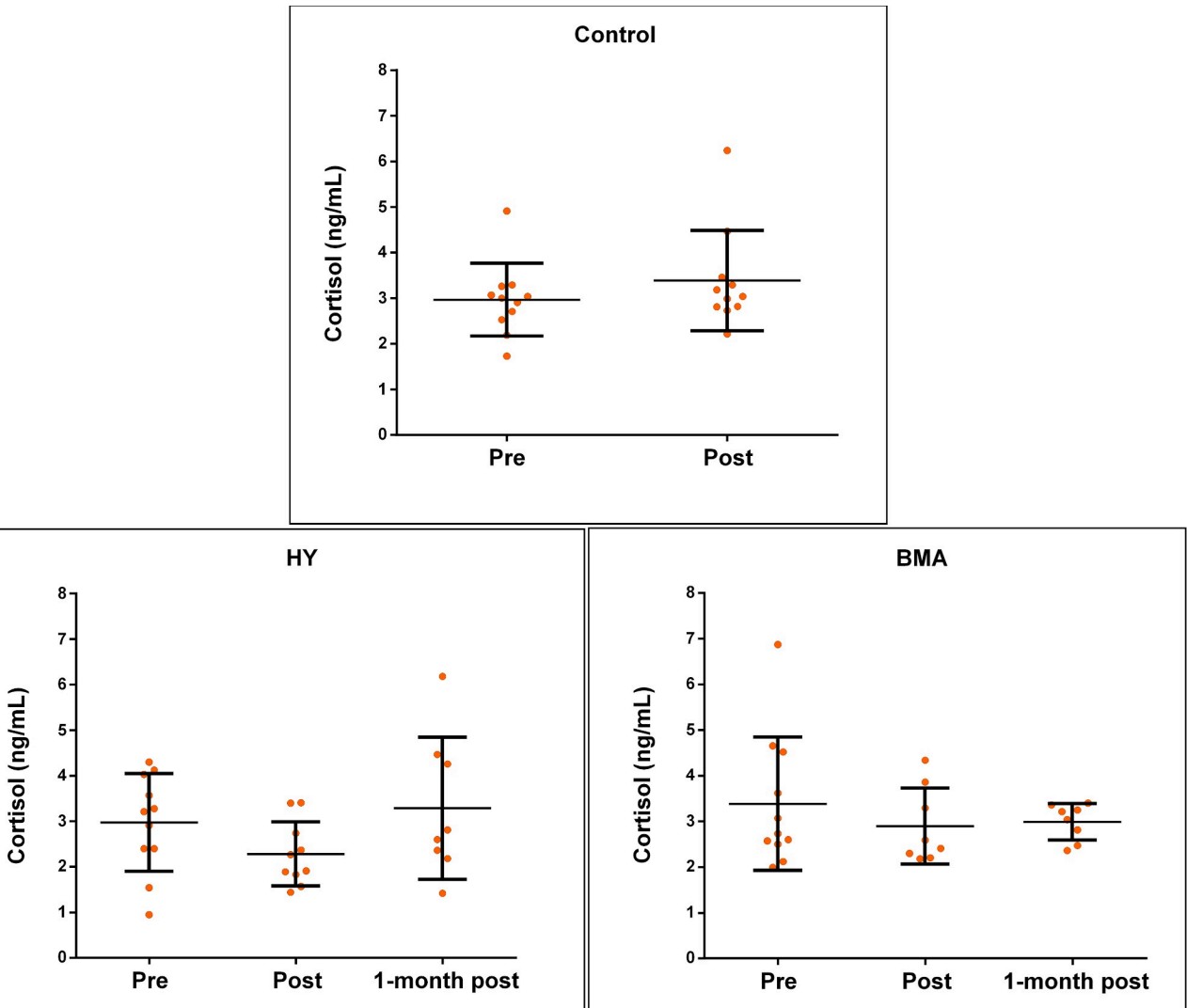

**Fig 7.** Mean and standard deviation of salivary cortisol level in: a) Control group, b) Yoga Hatha group (HY) and c) Body Movement Awareness group (BMA). Pre = before program start; Post = just after program end; 1 month post = after one month of program end.

the BMA group than in the HY group, and the values were grouped tighter around the mean. In the HY group, the mean value increased by more than 40%, and the range of values increased almost 2.5 times from when the intervention ended.

We also collected saliva samples in 3 sessions during the implementation of the program, before the beginning of the session and at the end of it (Fig 8). In the case of the HY group, although no significant differences were found, the effect size increased from sampling two to sampling three, reaching values > 0.5 in the third session. No effect was found during the first session. The opposite was observed for the BMA group, which showed a change in the effect size when comparing the levels of cortisol before and after session one (Cohen's d = 0.537).

## Content analysis of reflective diaries

We analyzed the data from the reflective diaries of 21 teachers belonging to the HY and BMA intervention groups. The deductive paradigm was applied for the interpretative analysis. We

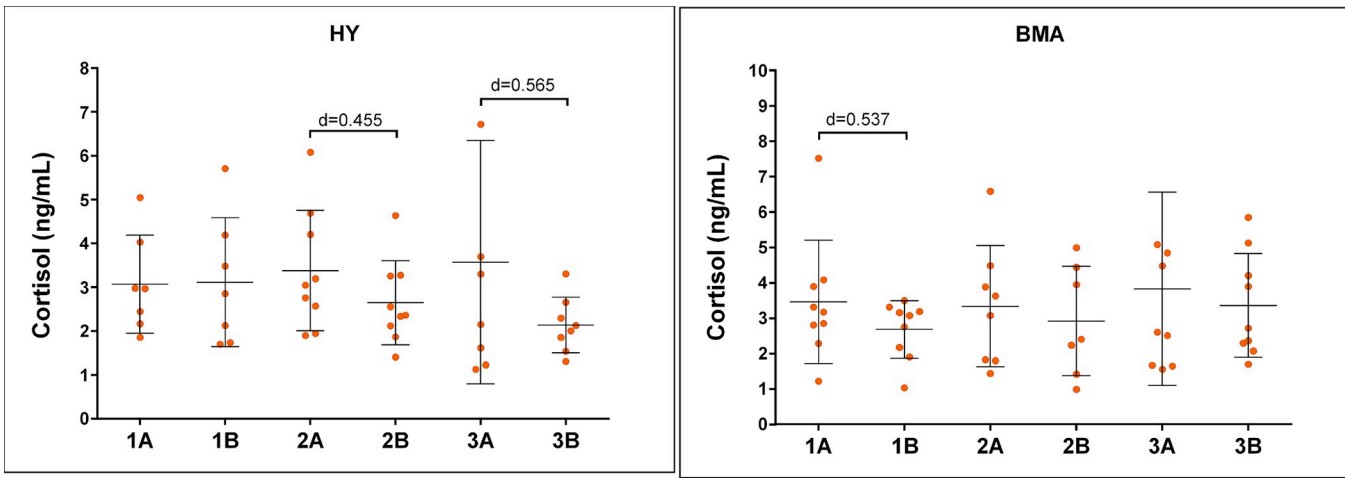

**Fig 8.** Salivary cortisol level for Hatha Yoga intervention (HY) and Body Movement Awareness intervention (BMA) groups over 3 different sessions (1–3) before session start (A), and after session end (B). The graph shows mean and its standard deviation; d-value represents Cohen's effect size.

used Mehling's categories (2009) for body awareness as a starting point for coding, although new categories arose in the analysis. Two large metacategories (domains) emerged corresponding to intrapersonal and interpersonal content (Table 6).

*Intrapersonal* refers to what occurs within a person's mind, including the introspective reflections looking inward and figuring out their own feelings, sensations, motivations and

**Table 6. Definition of structure and meaning of categories and sub-categories that emerged during content analysis of reflective diaries.**

| | INTRAPERSONAL DOMAIN | |
|---|---|---|
| **Acron** | **Categories** | **Definition and themes** |
| PBS | Perceived body sensations | Ability to notice changes in body processes, identify internal sensations and discern subtle body signals, indicating functional variation in the state of the body or its organs as well as its physiological and emotional state (relaxation/stress). It includes: a) Sensations of distress, worry, pain and tension (SD); b) Sensations of wellbeing (SW); c) Neutral or ambiguous sensations (NS); d) Affective aspect of sensation (AS). |
| AQ | Attention quality | A continuum ranging from active attention to sensations to distraction, ignorance, and suppression of perceptions. The focus of attention may be involuntarily reactive or intentional. It includes: a) Intensity (IN); b) Self-efficacy in attention control (SEAC); c) Mode: thinking/labeling vs. experiencing the present-moment (MD). |
| AT | Attitude | General bias in the activation and interpretation of sensations. Variation in how one relates one's bodily sensations. It includes: a) Trusting (T); b) Catastrophizing (C). |
| MBI | Mind-body integration | Subjective experience of oneself as a whole. It includes: a) Emotional awareness (EA); b) Overall felt sense of embodied self vs. feeling disconnected (OSE). |
| LE | Learning | Ability to incorporate new knowledge about one's own body, including surprising discoveries, and the body-mind relationship as well as its extrapolation or application to other contexts. |
| SP | Satisfaction with the program | Awareness of the program's positive contributions to oneself. Recognition of the own needs, surprise for being able to freely move and feel the body and enjoyable rememberings that emerge through the body. |
| | INTERPERSONAL DOMAIN | |
| CO | Confidence | The feeling that one can have faith in or rely upon someone or something. It includes: a) Trust on moving out of the own kinesphere; b) Interaction with others through movement. |
| PF | Play | It is an activity engaged for enjoyment, fun and healing. |
| PVC | Physical & visual contact | Physical touch and visual contact promote intimacy with others. An aspect of polarity appears between difficulties in giving and receiving contact, and the pleasure and knowledge that it brings. |
| QM | Movement qualities | Efforts and actions of Laban Movement Analysis related with one's own personality. |
| KE | Kinesthetic empathy | The ability to experience empathy merely by observing, and attunement to the movements of another human being. It encourages the ability to listen and adapt to others. |
| TK | Thankfulness | Feeling of being grateful to the innovative proposal and the creation of a self-care atmosphere. |

goals. The dimensions of Mehling et al. (2009) [71] were included in this metacategory as it refers only to self-centered contents––that is, physiological and emotional contents that do not require relationship with the environment, or it is even necessary to abstract from it. Two more categories, learning and satisfaction with the program, were added to those of Mehling's. However, from the enactive knowledge the research team found it necessary to add the meta-category domain *interpersonal* to the construct of body awareness, defined through relationships with others in the space, which includes the way in which one's own body perceives and answers to others. The need to add this category arose during the data analysis process because numerous manifestations emerged from the participants that could not be compiled into any

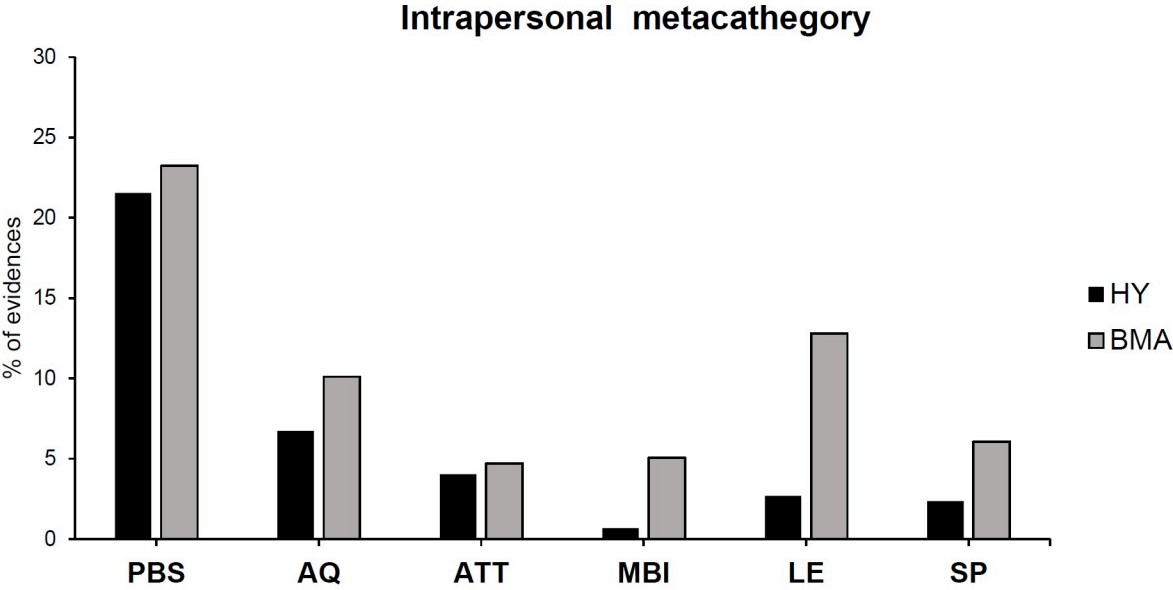

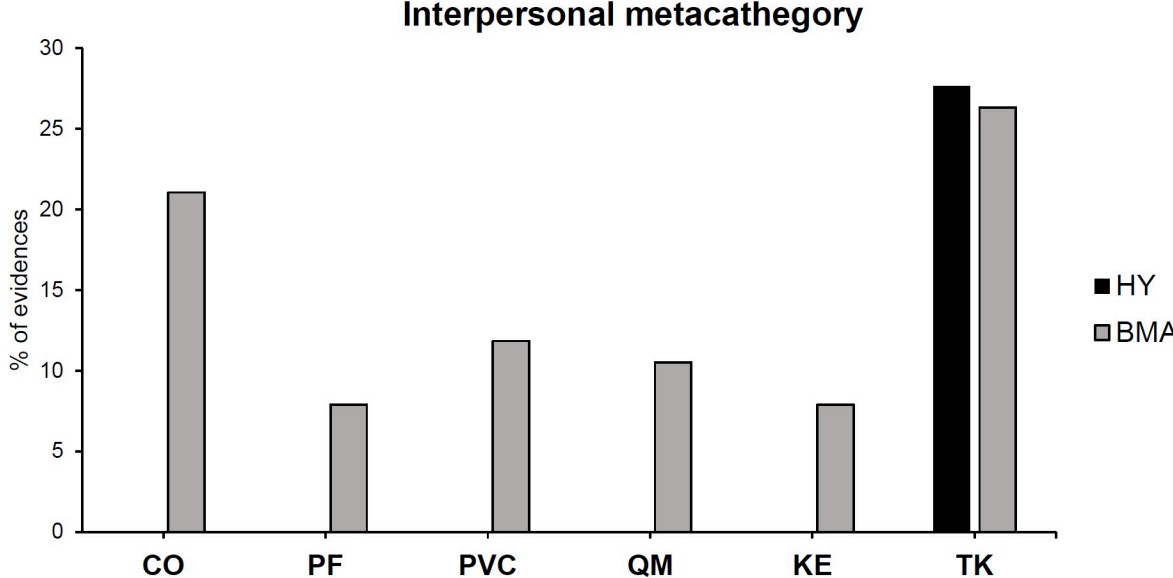

**Fig 9. Percentage of evidence for each category of intrapersonal and intrepersonal metacategories for both HY and BMA groups.**

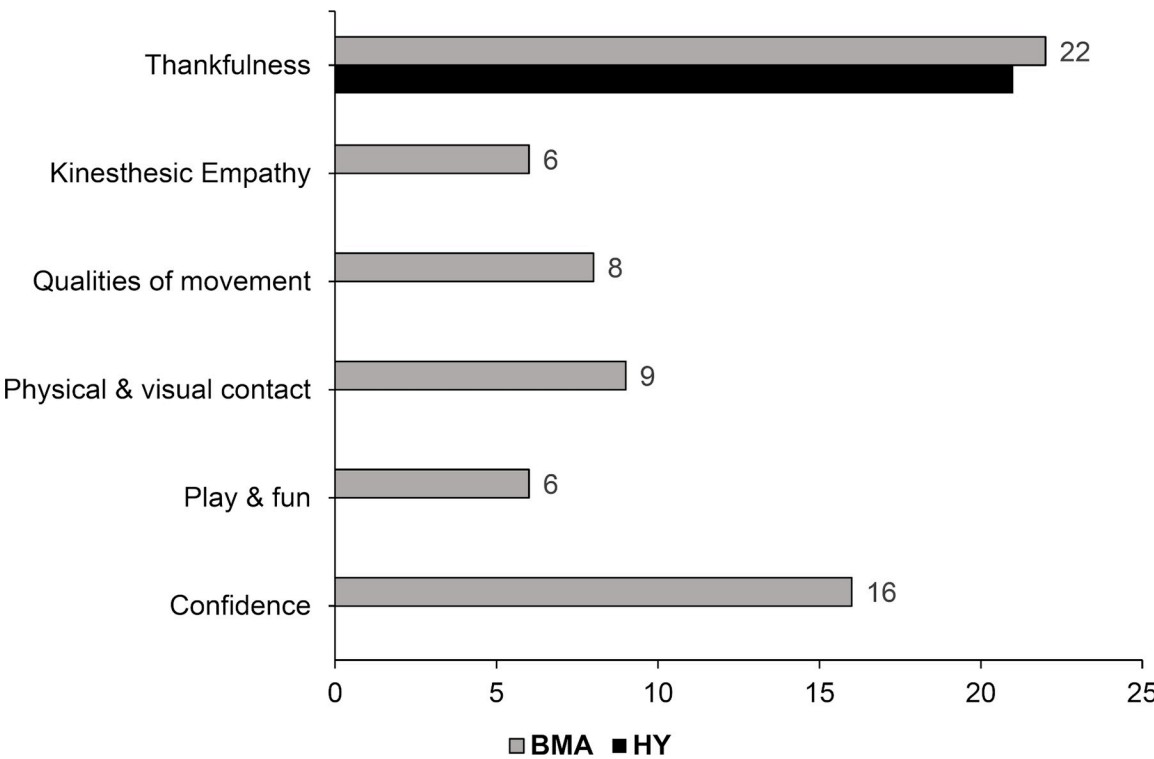

**Fig 10. Evidences for categories for interpersonal domain.**

other category or subcategory. Table 6 presents the complete structure of categories and Fig 11 lists the number of evidences for each category. Mehling's categories and themes for each category are shown in a shaded area in Table 6.

Fig 9 shows the percentage of evidence for each category of intrapersonal and interpersonal metacategories. With the exception of the Thankfulness (TK) category, for which there was a similar number of evidences in the HY (21) and BMA (20) groups, the remaining categories under the interpersonal metacategory show evidence contributed by the participants of the BMA group.

**Intrapersonal domain.** Six categories were established in this domain, as defined in Table 6. The total number of contributions from participants is shown in Fig 11. Some of the evidence collected is presented as an example.

*Body sensations perceived.* There was a wide repertoire of perceived body sensations. There was a clear difference between the number of evidences on sensations of discomfort, pain or tension, which appeared more frequently in the HY group (24) than in the BMA group (12). One participant wrote: "My body was a–scream-, everything was complaining when I stretched" [HY-P10-S1]; or "Today I noticed a lot of tension in the hamstrings, I had trouble stretching my quadriceps and also discomfort in my feet until I managed to find a posture of balance". [HY- P17-S3]. By contrast, sensations of wellbeing appeared more frequently in the BMA group (30) than in the HY group (23). These sensations of wellbeing increased as the sessions progressed, "Once incorporated into the group dynamic, the sensation of relaxation has been greater at the end. The perception of effort in the postures has also decreased" [HY-P3-S3] or "As always, I am fascinated that in less than ten minutes in the final relaxation I feel like I want to stay like this forever, super relaxed" [BMA-P5-S8]. Some of the evidence was ambiguous or neutral in relation to the sensations, such as "It is time to give and receive a

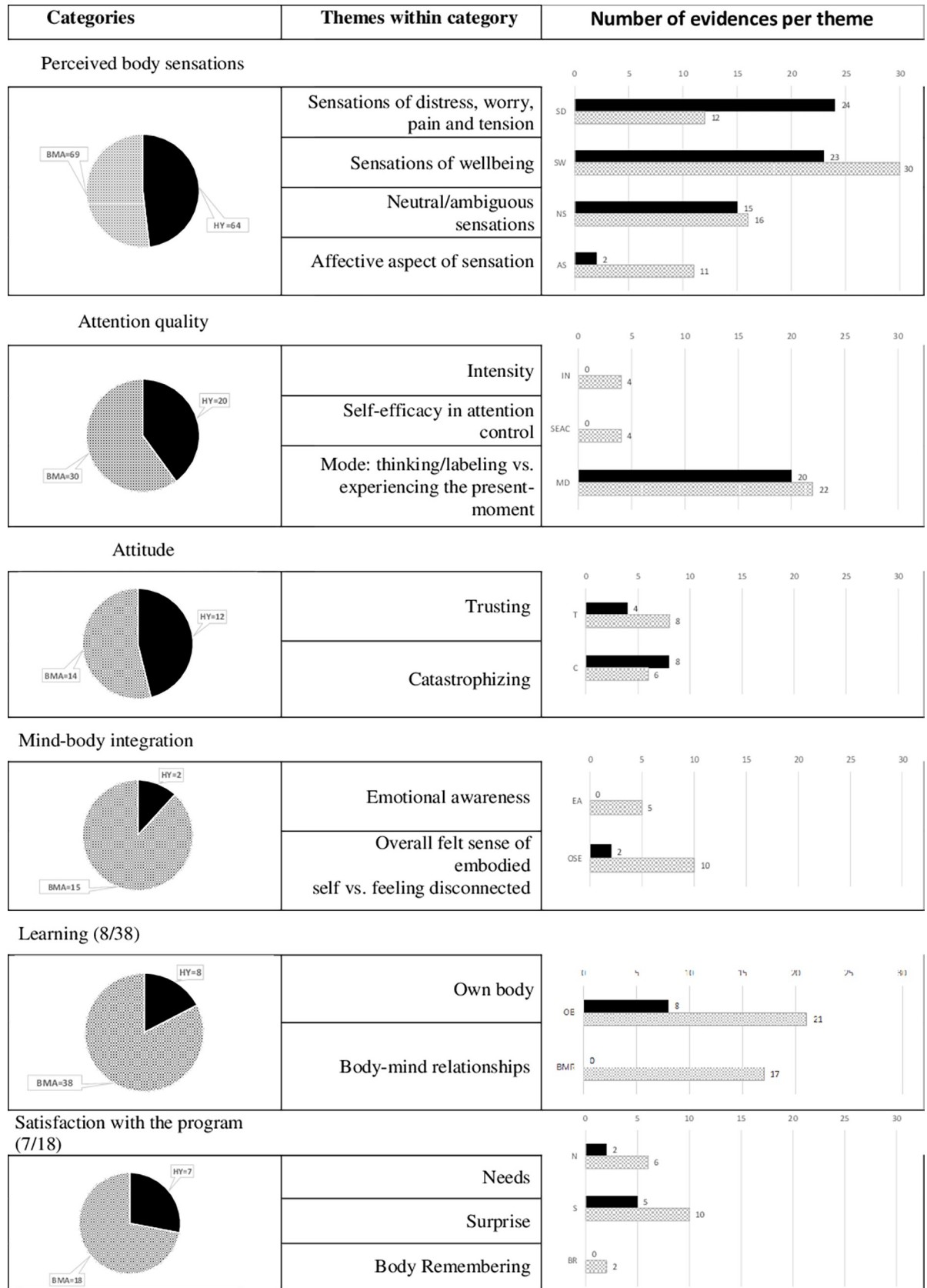

**Fig 11. Evidence for categories and sub-categories for the intrapersonal domain.** Bars and portions in black represent the number and percentage of evidence for the BMA program, while bars and portions with lattice represent the number and percentage of evidence for the HY program.

massage (skin, muscle and bone). I allow myself to be here, feeling what I am receiving and what I am giving [BMA-P5-S4] or "I noticed muscles pain but as a warning that they were alive, not as something negative [HY-P10-S2], respectively. As for the affective aspect of sensation, almost all the evidence appeared in the BMA group, "This session has served to reduce my fear of touching. I am relaxed about being touched, but I have respect for the possibility of offending someone by touching them. However, the last few sessions have taught me that, in the appropriate context, we all feel good about physical contact" [BMA-P8-S4]. Another participant said, "The feeling of letting the body do what it wants without thinking is very liberating. The body knows what is right for it. There comes a time when the body acts alone, without the need for any conscious guidance. All the muscles lose their superfluous tension and simply move the body as they wish" [BMA-P11-S5].

*Attention quality*. The intensity of the quality of attention changes; participants increase their capacity to pay attention, as "I have been awakening the awareness of the different parts of my body and my senses" [BMA -P9- S3]. Likewise, efficiency appears in terms of attention control, "Awareness of each joint, of each possibility, of each option" [BMA-P25-S6]. Participants begin to be able to allow themselves to explore and be in the present moment without thinking or labeling (attention quality mode):" The body knows at every moment what posture or action is good for it without the mind interfering" [BMA-P11-S5], but they also realize when thinking interferes with the present moment "I need to stop thinking! During the moving presentation, I find myself thinking too much" [BMA-P8-S2].

*Attitude*. There was a more catastrophist *versus* trusting way to see the own experience in the HY group (8/4) in comparison with the BMA group (6/8). A participant who had noticed an impact on how he did things in a previous session said, "But these changes only lasted that day. The next day I was feeling the pressure of having too much to do and not enough time [HY-P4-S4]" or "I have a hard time doing those exercises where the lack of flexibility prevents me from noticing the changes in my body. Actually, what I do is be aware of what my body is struggling with" [BMA-P21-S6].

*Mind-body integration*. There was an awareness of emotions in relation to one's body "I am ashamed of my body" [BMA-P5-S1], and also feelings of connection or disconnection in relation". There was a need for integration "by becoming aware of each part of the body, it is something I should do every day to keep my mind in shape and to release what I am accumulating every day, every week, every month". [BMC-P11-S3]. Another person goes as far as to say, "I liked the exercise in pairs, but I didn't feel in my element (the body is not my thing)" [BMA-P5-S1]. He talked about his body as if it were not his own. Another participant found a connection with what happens outside the room "I have no trouble being aware of my joints and being able to look for movement options when I have a limitation in a joint. I see it as a simile to life: if we encounter a difficulty, we must look for other options that satisfy us" [BMA-P9-S6].

The following two categories were added to those found by Mehling et al., (2011) [34]:

*Learning*. Many participants mentioned the learning that they experience during the process. Sometimes the learning is directly related with the own body (limits, more predominant movements, . . .): "It was curious to see how little we know about our bodies, that we did not identify several of the bones [BMA-P6-S4] or "I didn't imagine that the same movement could have so many interpretations" [BMA-P5-S13]. In other occasions, the learning was achieved

through the body-mind experience and applied to their own personal development. One participant mentioned: "Once again, as in all these workshops, there was something (or a lot) that helped me to analyze, channel and manage the current personal moment in other ways" [BMA-P1-S4]. Others shared: "I didn't know how to ask for help, I didn't know I needed it" [BMA-P5-S4] or "You can work with your own strength by investing it for your own benefit or to support others, but sometimes you can also use the strength of others to let yourself go. And that sometimes releases the burden and moves you forward. Great learning" [BMA-P11-S3].

Notably, the learning evidences were more frequent in the BMA group (38) than in the HY group (8).

*Personal satisfaction.* Learning goes hand-in-hand with great personal satisfaction, which was, again, more evident in the BMA group because of the greater number of evidences (18 vs. 7 in the HY group). The participants mentioned how gratifying it was to allow time for listening to one's own <u>needs</u>: "all these sessions have been very important because they motivate us to have and to take care of our space and to cultivate that inner part that we sometimes forget" [BMA-P22-S4]. A participant talked about the needs in a very different way, gaining insight: "Becoming aware of my body has helped me to get out of it, to get rid of it and to be able to think more clearly. Thinking and coming to the conclusion that what I have to eliminate, what I have to get rid of, is the needs" [BMA-P11-S3].

There were also many moments of <u>surprise and joy</u> at the discoveries: "How difficult it is to move slowly and let yourself go, and how good it feels when you manage to do so! Why do we stop playing when we are adults? [BMA-P25 -S5], or "My body wakes up by performing movements I don't usually do every day. It gave me a feeling of freedom and joy" [BMA-P22-S2]. Another participant was touched by the fact of being able to dance: "There was a very surprising moment: when the guide told us that we were dancing! Perhaps my movements are not as direct as I thought" [BMA-P25-S2].

There were also moments when the <u>body memory</u> came out: "I began to remember that I have a body that works and can move in different ways" [BMA-P22-S2]. Another participant discussed a very intimate and personal moment of remembering in the space: "The posture of the "cat" reminded me of the birth preparation courses and I felt this mixture of nostalgia and relief" [BMA-P27-S2].

**Interpersonal domain.** The interpersonal domain included testimonies that show aspects related to relationships with others and the way in which the person perceives the environment and how they respond to it. The definitions of these categories and the number of evidences are shown in Table 6, Figs 10 and 11, respectively. Most of the evidence (76%) came from the BMA group covering six different categories, and 24% came from the HY group. In the latter case, all the evidence was under only one category: **Thankfulness**. In fact, participants of both groups expressed a lot of gratitude and thankfulness during all the process, but especially in the last session. They thanked the work atmosphere, the tools, professionalism and the time offered, "I want to thank [session leader's name] for creating such a pleasant climate in the session, so that we all felt comfortable moving and expressing ourselves with our bodies. It made us really feel the moment. Thank you!" [BMA-P5-S8], or "Thank you for the tools provided, for allowing us to spend time on our wellbeing. It has been very rewarding and really short" [HY-P28-S8].

The other five categories were determined under this domain, with evidence only in the BMA group:

**Play and fun:** The participants perceived the sessions as a space for play and fun, which increased the confidence in the space and in the group. One of them mentioned: "The final part was very funny and I really felt like I was dancing and I never do! "[BMA-P8-S4]. They

also mention the importance of playing: "Playing is healing for the body and the mind: I go home with this sensation" [BMA-P22-S5].

**Confidence**: Participants freely expressed difficulties related to the work through a channel other than verbal, but at the same time there was a progressive acquisition of confidence in the group, allowing them to show themselves in an authentic way through the expressive movement: "Interaction with peers was more complicated at first but we quickly started to interact" [BMA-P22-S1]. Some testimonies were touching: "When a person can get out of himself and listen to the other there is a possibility of personal growth. The lesson learned today: listening to the outside world. Getting out of myself, synchronizing with others, seeking collaboration in order to grow and create. This is really important for me now" [BMA-P11-S8].

**Physical and visual contact**: This category is connected to the previous one, because physical contact can easily emerge if there is confidence and respect in the group. One participant said "when we started to move including the physical contact it seemed that we were afraid to move too much. Little by little, we became more confident and it was less difficult for us to communicate" [BMAA-P13-S4]. The direct visual contact was experienced as a difficulty for some of the participants: "I found that direct visual contact made me shy so I avoided it. I felt more comfortable with my eyes closed" [BMA-P5-S4]. Contact was perceived as something nutritive by some: "How much we need contact with others! It makes us aware of our body and the body of the other" [BMA-P6-S3]. The participants recognized the importance of physical contact and the prejudices they have developed "Today I have been fully aware of the little physical contact I have in my day-to-day life. That I touch people little and that I am touched little too. That conscious physical contact is another way of communicating that, in my case, I hardly use. It seems to be reserved for games with children that, when you deal with adults, "that is not done" [BMA-P8-S8]. **Movement qualities**. The transition from the intrapersonal to the interpersonal domain can be seen in the use of outer space (beyond the kinesphere itself) and in the dynamics of interaction between the participants. The use of specific tools of movement analysis such as those proposed by Laban provide the participant with greater knowledge of their way of relating to the environment; for example, "I feel the sudden movements as aggression, I've noticed that some people in the group have had a harder time with this speed, me included. I was afraid of hurting someone" [BMA-P9-S2]. Some participants understood that they can self-regulate their own energy and strength in relationships with others, "You can work with your own strength by investing it for your own benefit or to support others, but sometimes you can also use the strength of others to let yourself go. And that sometimes releases the burden and moves you forward. Great learning" [BMA-P11-S5]. In addition, as sessions progressed, they were able to relate their movement patterns to their behavioral modes, "By reflecting on how I was doing the exercises in a direct or indirect way, I think that at work or in my duties I like to behave in a direct way but in my leisure time I prefer to be indirect. This in turn leads to greater self-knowledge" [BMA-P4-S7]. Thus, both domains, the intrapersonal and the interpersonal, are continuously related and one nourishes the other in the process of personal growth. **Kinesthetic empathy**: the process of empathizing through body experience was something new for the participants, and in fact, difficulties did arise: "I'm left with that moment when I couldn't tell if I was imitating my partner's movements or the other way around. Unfortunately, we didn't know how to stay in tune when we opened our eyes and the connection was lost" [BMA-P25-S8]. "At the same time, the ability to listen to the other person and adapt the answer through a non-verbal expression: Today I have been able to adapt my energy to the strength of the other person. I can listen to the other's body and adapt to its possibilities" [BMA-P9-S5]. This activity of listening leads to a place of great connection with the partner: "It was a session of communion and connection with the other on a corporal level. The most striking sensation, shared with my partner, was the first part: after a long

period of mobility on both sides, we did not know who was initiating—leading—and who was accompanying, a very positive communicative experience" [BMA-P26-S6].

## Discussion

The aim of this pilot study was to analyze and understand the impact and contributions that two different body-mind training programs could offer in relation to subjective well-being, stress level and degree of self-knowledge of university teachers. A mixed methodological approach was implemented following guidelines that propose its presence in the field of health. This approach makes it possible to complement the information obtained from a quantitative approach in studies in which the sample size cannot be large due to the characteristics of the interventions and relational component is an important one. But it also makes it possible to carry out an in-depth reflection on the experience of the participants, their thoughts, emotions and sensations, beyond what the numbers can say [73].

Post-measurements coincided temporally with the examination period. The literature shows that the level of stress increases not only for students but also for teachers [74–76]. Both groups showed reduction in stress at the end of the intervention with respect to the increase expected due to the arrival of the exams, significant for the HY group and a tendency for the BMA one. Other studies have also shown the effectiveness of Hatha Yoga and BMA in reducing stress levels [46, 47, 77]. This was corroborated by the increase in subjective wellbeing and body awareness, also significative for HY group. The superior results in the HY group over the BMA group in the BAQ test might reflect improvements related to aspects connected with proprioception and interoception [71, 78]. A tendency, yet non-significant, development was observed in the BMA intervention group.

It is difficult to explain why an intervention that promotes increased body awareness and movement does not score more strongly in the BAQ scale. This could indicate that the BAQ is not entirely useful for measuring changes following a BMA-type intervention. In this type of intervention, in addition to the proprioceptive and interoceptive aspects, the connection between bodily sensations and the sense of self, interpretation of external stimuli, emotional awareness and the integration of mental, emotional and physical processes are considered. The intervention therefore employs a broader conceptualization of the body awareness construct as defined by Mehling [71]. However, the BAQ assesses only one dimension of body awareness, that of perceived bodily sensations [71]. Future studies may follow the suggestion of Danner [79] and use body awareness questionnaires that do not completely ignore these dimensions that are important in a BMA-type intervention. Accordingly, it might be better to employ the BAQ tool together with others such as the Body Consciousness Questionnaire [80] or the Awareness-Body-Chart (ABC) [79]. We believe that something similar is happening with FFMQ. Neither of the two interventions was a mindfulness-type intervention, so the questionnaire fails to discriminate changes at this level. In fact, although there seems to be a relationship between mindfulness and body awareness, the results obtained so far are very diverse and do not make it clear what this relationship exists. The lack of evidence of this link may imply that actual increases in body awareness are not related to the benefits of mindfulness [81].

There was an increased in satisfaction with life in both groups after the intervention, significant for the BMA group. This is coherent with other studies which introduce creative movement and play in the interventions which increase vitality and feelings of joy and life satisfaction [82]. Participants' reflections gave complementary support to this aspect with many evidences about personal satisfaction, joy and surprise.

A high HRV appears to be an indicator of good health, lower morbidity and mortality [83], and can also be a useful biomarker in stress management [84] or emotion regulation [85].

In our study, LF power and LF/HF ratio increased whereas HF power decreased. When the frequency domain is used for HRV evaluation, the LF/HF ratio can estimate the relationship between sympathetic and parasympathetic nervous system activity under controlled conditions, where a decrease in the LF/HF index reflects parasympathetic dominance, whereas an increase indicates sympathetic dominance [86]. Although these results might seem contradictory to what is expected when seeking to develop vagal control by applying body-mind interventions, many authors have obtained the same profile of LF increase and HF decrease using different approaches such as Zen meditation [87], or with yogic practices to slow down [88] or retain [89] intermittent breathing. Hewett et al. [90] also failed to increase HF through a 16-week program of Bikram Yoga (90 min per week). In the case of a HY intervention, the parameter LF/HF increased significantly [91], as in a previous study by us using Vipassana meditation [92].

The aforementioned studies using body-mind approaches that failed to increase HRV do not contradict the fact that such interventions are effective in reducing psychological stress [93, 94] and in controlling stress levels [95].

Our findings for the HY and BMA interventions were very similar in significance and effect size, making it difficult to differentiate between them for their impact on HRV. This led us to reflect on what aspects both interventions shared. One of the requirements for inclusion was that potential participants should not have a lengthy practice with any of the tools that would be proposed in the program. Thus, all participants, irrespective of the group to which they were assigned, had to actively practice their attention following the instructions of the facilitators. This could be the reason for the activation of their sympathetic system similarly to what was reported by Telles et al. [96, 97] in a comparison with different types of mediation. It is possible that the interventions triggered an activation of the sympathetic nervous system due to the attention span promotion by body-mind interventions, but also a vagal activation that reduces stress as referred to by other studies [93–95].

The results obtained in the present study show that variations of saliva cortisol showed a trend in favor of its use as a measurement of stress [98–100]. Even opposing results can be obtained depending on how demanding the intervention is [101–108]. In the case of the present study, the trend-level findings suggest that there is a moderate effect on cortisol levels when university teachers participate in a program that helps them come into contact with their body, in comparison with a control group that does nothing. There is a reduction in both groups after the intervention and BMA appears to have a greater "memory effect" after some time has passed from the end of the program.

The inability to find significant differences in cortisol might be attributed to the fact that three differentiated behaviors were observed in the HY and BMA groups. Some teachers increased their cortisol levels as the term progressed in a manner consistent with the expected growth in work-related stress, whereas other participants showed no variation, and a third group (rarer) showed decreased levels, going against what would be expected. These patterns did not seem to be related to the group to which they belonged but seemed to be characteristic of the individual analyzed. The co-existence within the same group of these opposing patterns could be responsible for the lack of significant differences between groups. Hakamata et al. (2017) [107] suggest that this difference in behavior could lie in the amygdala functional connectivity, for which they found different patterns.

The reflective diaries provided abundant evidence to complement the quantitative results and guide the future search for better tools. In relation to this, the first aspect that should be mentioned is that the profusion of reflections was much greater in the BMA group (251, 65%)

than in the HY group (134, 34,8%). We believe there are several reasons for this. The Yoga program sought to repeat some postures or asanas, remaining constant throughout all the sessions, with the first session introducing the pattern of exercises. By contrast, in the BMA program, the first session was dedicated almost exclusively to generate the necessary confidence in the group, through dynamics of game and of progressive contact that allowed the establishment of interpersonal bonds, and was later deepened with more facility in experiential aspects. The process of internalization and awareness is of greater depth and complexity in the BMA approach. One of the reasons for this is that the HY design was oriented towards a work of introspection, of listening to one's own body. This had an impact on reducing the stress levels of the participants, who also experienced relaxation and a sense of wellbeing after each session, as corroborated by other studies [93, 102]. The structure offered and the repetition of almost identical asanas in all the sessions allows the participant to learn the movements, so the attention can be shifted from the need to memorize the movements to move towards one's own sensations and perceptions. This is further corroborated by the results of the PSS, which showed a significant decrease in the HY group. This structured framework, as offered by other mindfulness-based approaches, helps participants to increase body knowledge, body awareness and self-confidence regarding the own body [30].

The BMA group combined time for introspection with time for creative and expressive dynamics in a context of group interaction. The use of creative resources has been widely used in the field of mental health [47, 101, 109, 110]. Its potential has, however, not been explored to any great extent in university teacher training [111]. The impact of creative approaches on personal wellbeing and stress reduction is well known [44, 47]. In our study, only life satisfaction had a tendency to be significant in the quantitative analysis but changes in wellbeing and stress reductions could also be observed. The respective artistic media thereby provide different methods to activate resources and coping abilities and increase action flexibility, self-efficacy, and empowerment [47]. Even though the structure of the session was always the same, the theme worked on changed in each, with different activities and proposals that continuously forced the participants to step outside their comfort zone with the difficulties that this entailed. In a changing world, where teachers are asked to move towards transformative education and create challenging creative spaces for the students, it is important that they also have the experience of new ways of learning [42, 111, 112]. The participants in our study had to be open to uncertainty and transformation, which are also characteristics of their teaching-learning context [113]. Without being a therapeutic work group, undoubtedly working with the participants' own movement repertoire makes them aware of themselves and expands the possibilities of movement through new experiences [54, 114]. In both groups, cortisol levels decreased at the end of the program compared with the control group, and this was also maintained one month after the end of the intervention in the BMA group. Although this decrease was not statistically significant, the numerous mentions of the participants revealed an impact on their state of stress and wellbeing.

Another differentiating aspect between the two approaches was the relational aspect. From early childhood, a person becomes aware of themselves and their place in the world through their relationship with others, first and foremost a fundamental relationship with their mother [115–117]. This is produced through non-verbal channels and the embodied experience [36]. In our opinion, it is not possible to speak accurately of bodily awareness excluding the relational mechanisms. The use of space, the form and the corporal tone that our body adopts before the different external stimuli and the qualities of expressive movement are characteristics that modulate the relation with the other [40]. Qualitative exploration has captured these relational aspects that are essential in the individual's state of satisfaction, well-being and self-knowledge and that should be taken into consideration in the educational setting in which the

teaching-learning process is based on the teacher-student relationship. This suggests the need to incorporate other types of data collection tools in future feasibility studies. Some of the quantitative instruments used in this project seem to be more appropriate for the HY approach.

Thus, as the richness in the HY approach was in the intrapersonal domain, a movement of transition from the intrapersonal to the interpersonal and back was continuously occurring in the BMA group. This has parallels with the transition from the non-verbal to the verbal. Some participants had difficulties in putting the experiences into words, but at the same time the fullness of responses that emerged shows the need to anchor what is generated in movement [118, 119]. There were moments of true insight, of appreciation of unconscious contents that came to light and gave the participant a possibility of integration and acceptance. Furthermore, there were opportunities for playing that were valued by the participants. Playing, which cannot be dissociated from creativity and a sense of "enjoyment", is an "intensely real" experience that has intrinsic therapeutic virtue, that is to say it is capable of promoting "self-healing" [115–120]. According to Winnicott, it is necessary to establish a positive social attitude towards playing. Participants agreed this idea. The combination of both approaches could benefit and be enriched by the specific differences and contributions observed. Thus, an intervention that, for example, combined throughout the session the regular and repeated structure of a set of asanas together with relational dynamics of exploration through creative movement could contribute to a decrease in stress and an increase in well-being, while incorporating learning that arises in a relational context. This could contribute to more healthy environments in higher education as other studies have proved [121].

## Limitations and future directions

Some important limitations should be taken into account. As other authors have suggested, the environment in which the experiment is conducted––including the reasons why participants choose to participate––may influence the adaptations of physiological stress markers [90]. This is clearly a limitation of our study that must be highlighted. All participants were volunteers. This is a potential bias but, on the other hand, by the very nature of the programs, it would not make sense for the university to force its faculty to join them. Another limitation was the small sample in this study (n = 31) that prevents to generalize results. We think that some of the tendencies observed could be confirmed with a bigger sample size. For future studies it will be necessary to first characterize the physiological patterns of cortisol characteristic for each individual and use this information as an inclusion criteria. It remains also to be considered if the sampling time was the most appropriate. *A priori* it was so, because after midday cortisol levels stabilize from the moment of awakening in which they rise and then drop. However, a recent study that investigated the use the cortisol in saliva for clinical diagnosis and monitoring of burnout found that the best time to compare the levels with a control group is at noon or bedtime, showing a superior diagnostic capability [108].

The requirement that people delivering the two interventions were experts with long experience and background meant that two different instructors were required. Although we do not consider that this may have influenced the results, it is something to take into consideration for future studies with a single expert carrying out both interventions.

Other contextual factors also need to be taken into consideration. Proposals such as the current one require a suitable and comfortable physical space, with natural light that allows the intervention to be carried out without interruptions and guarantees the necessary intimacy and privacy. The timetable for the activity is not a minor element, and should be carefully chosen according to the objectives pursued. At the same time, it may adjust to the requirements of

the institution in terms of lessons schedule, break times, etc. In our experience, it is important to have enough transition time for arrival and departure, so the activity can be carried out under the best conditions. All these aspects undoubtedly require a commitment on the part of the institution to invest time and resources in promoting experiences that can contribute to generating healthier educational spaces.

Furthermore, the choice of assessment instruments seems to be more appropriate for the HY approach. Qualitative exploration has captured relational aspects that are essential in the individual's state of satisfaction, well-being and self-knowledge and that should be taken into consideration in the educational setting in which the teaching-learning process is based on the teacher-student relationship. This suggests the need to incorporate other types of data collection tools in future feasibility studies.

## Conclusion

Body-mind approaches have been proven useful in reducing stress in higher education teachers. The HY program had a more pronounced acute impact on the parameters of perceived stress, body awareness and subjective wellbeing, while the BMA program produced high levels of life satisfaction, was able to sustain the reduction in cortisol levels longer after the intervention was completed, and to produce evidence of self-awareness, self-knowledge and social interaction outcomes. Contrary to what was expected, the decrease in stress levels was accompanied by an activation of the sympathetic nervous system, observed through analysis of HRV. This activation would be justified by the need for care capacity development in both interventions with respect to the control group.

In spite of the limitations of this study with respect to the small number of participants, we believe that it has particular relevance because it is a pioneering study in this population. It will, however, be necessary to continue to reinforce the tools, both physiological and psychometric, so that the results are more robust. It will also be necessary to refine the tools that evaluate body awareness, to allow us to capture all the nuances of the body awareness construct when using BMA-type interventions. The HY program has proven to be a reference program that can be used in any type of intervention that seeks to reduce perceived stress or improve wellbeing among university professors. Regarding physiological tools, future research should: 1) characterize the basal cortisol levels before the beginning of the study and use them to configure the groups; 2) demonstrate if sampling saliva later (noon or bedtime) has any implication in the reliability of cortisol results; and 3) verify that the body-mind programs in higher education teachers produce the non-expected HRV behavior. Both approaches offer aspects of interest and have been extensively applied with good results in mental health environments. HY has a very structured framework where attention is oriented to one's own bodily sensations and bodily limits. The BMA approach introduces creativity, movement profile analysis and group work in a relational context to increase self-awareness and self-knowledge, and develop communication and kinesthetic empathy. In our opinion, a combination of both programs could provide a safe framework that would help to increase the levels of teacher wellbeing and self-knowledge. This could also contribute to improve the teaching-learning processes by generating resources that improve presence and communication with students.

## Supporting information

**S1 Appendix. Procedure to translate to Spanish the Body Awareness Questionnaire (BAQ) (Shields, Mallory & Simon, 1989).**
(DOCX)

**S2 Appendix. Requirements and limitations for participants' saliva collection for the determination of cortisol level.**
(DOCX)

**S3 Appendix. Additional statistical data analysis.** ANOVA post intervention and paired-sample t-test pre and post intervention for all the variables and groups.
(DOCX)

## Acknowledgments

The authors wish to thank to Kenneth McCreath for proofreading the manuscript and all the participants for their compromise and participation. They also thank to Gloria Alvárez-Losada for her leadership and kindness during the HY program.

## Author Contributions

**Conceptualization:** Rosa-María Rodríguez-Jiménez, Manuel Carmona.

**Data curation:** Sonia García-Merino, Begoña Díaz-Rivas, Israel J. Thuissard-Vasallo.

**Formal analysis:** Rosa-María Rodríguez-Jiménez.

**Funding acquisition:** Rosa-María Rodríguez-Jiménez, Manuel Carmona.

**Investigation:** Rosa-María Rodríguez-Jiménez, Manuel Carmona.

**Methodology:** Rosa-María Rodríguez-Jiménez.

**Project administration:** Rosa-María Rodríguez-Jiménez.

**Resources:** Manuel Carmona.

**Software:** Sonia García-Merino, Israel J. Thuissard-Vasallo.

**Visualization:** Israel J. Thuissard-Vasallo.

**Writing – original draft:** Rosa-María Rodríguez-Jiménez, Manuel Carmona, Begoña Díaz-Rivas.

**Writing – review & editing:** Rosa-María Rodríguez-Jiménez, Manuel Carmona, Sonia García-Merino.

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
