## [Decision Letter · Decision Letter 0]

15 Jun 2022

PONE-D-21-31772

Stress, subjective wellbeing and self-knowledge in higher education teachers: a proposal for intervention through bodyfulness approaches

PLOS ONE

Dear Dr. Rodríguez-Jiménez,

Thank you for submitting your manuscript to PLOS ONE. After careful consideration, we feel that it has merit but does not fully meet PLOS ONE’s publication criteria as it currently stands. Therefore, we invite you to submit a revised version of the manuscript that addresses the points raised during the review process.

Based on the reviewer's comments, provided in full within this email, please shorten and re-structure the Discussion section, generally reduce wordiness, combine the outcomes of the qualitative and quantitative approaches, and address the study limitations within the manuscript. Furthermore, the manuscript would also benefit from separately reporting the parts that relevant to the ‘development’ of the intervention, and the elements that involve the ‘delivery (see comment reviewer 1) 

We look forward to receiving your revised manuscript.

Kind regards,

Katrien Janin, PhD

Staff Editor

PLOS ONE

Journal Requirements:

2. Peer review at PLOS ONE is not double-blinded (https://journals.plos.org/plosone/s/editorial-and-peer-review-process). For this reason, authors should include in the revised manuscript all the information removed for blind review.

Reviewers' comments:

Reviewer's Responses to Questions

**Comments to the Author**

1. Is the manuscript technically sound, and do the data support the conclusions?

Reviewer #1: Yes

Reviewer #2: Partly

2. Has the statistical analysis been performed appropriately and rigorously? 

Reviewer #1: Yes

Reviewer #2: No

3. Have the authors made all data underlying the findings in their manuscript fully available?

Reviewer #1: Yes

Reviewer #2: No

4. Is the manuscript presented in an intelligible fashion and written in standard English?

Reviewer #1: Yes

Reviewer #2: Yes

5. Review Comments to the Author

Reviewer #1: It was a pleasure reading the authors’ work and in their detailed description of their intervention. However, the manuscript requires further re-writing.

-First of all, the authors need to provide a shorter and better structured Discussion section. Currently the section is more than 10-pages long and parts of it may be suitable for a dissertation chapter but offer little value to the discussion of the findings of their paper. For example, their opening paragraphs (lines 669-687) in their Discussion focus on their methodology rationale that would be a better bit to their methodology section. Similarly, the first few pages in the Discussion they go on to be a presentation of methodology-related relevant literature that may again be a good fit to a dissertation’s methodology chapter but it offers little to the Discussion of the research findings. My advice to the authors is to reduce significantly the length of that section. They can start with a couple of paragraphs discussing their research objectives and methodology decisions within the scope of relevant literature and the move on to discussing their findings in relation to other studies in teachers or application of these interventions in other populations

-The title of the manuscript mentions that this is a “proposal of intervention”, however their methodology, and results section describe what may be best defined as a “pilot” study. If the authors indeed aimed to report the development of an intervention, then they’d need t restructure the paper to clearly show what parts are relevant to the “development” of the intervention and what elements involve its “delivery”.

Reviewer #2: The current study has adopted a mixed method approach – using experimental data and content analysis – to study the efficacy of two intervention on stress, subjective well-being, and self knowledge of higher education teachers. The authors have wisely opted for mixed method considering the limitation of the study with only quantitative assessment, but effort is needed to combine the outcomes of quantitative and qualitative domains and present it as complimentary to each other. Although the attempt was elaborate with many variables, the study apparently has many limitations that need to be addressed before publication. Especially the variables described at different parts of the manuscript are not congruent. Eg. The study measures listed contain FFMQ, to assess mindfulness, but nowhere in the manuscript the authors have mentioned about mindfulness. Similarly, the variables measured and variables described as the outcomes are not the same. Further comments are added here.

Title: Why there are two titles given (1-6)?

Abstract: Wordiness can be reduced in the abstract. Eg. in lines 20-22, the word ‘theirs’ is repeated five times, which is unnecessary.

Clearly state the number of groups, ie. two experimental groups and one control group with number of participants in each.

Introduction: In line 57, please include the in-text citation apart from the numbering.

Materials and Methods: lines 163-166 – Please explain what PARQ+ did assess. “participants should 166 not have lengthy practice with any of the tools proposed in the program” – was this another exclusion criterion? (Then better to mention exclusion criteria) Does it mean that the participants should not already have lengthy practices? Some more clarity is required here as what the authors have meant, since the intervention is not elaborated in the manuscript so far. Also please define lengthy. What was the duration beyond which a practice was considered lengthy?

Lines 171-173: please mention the number of participants in all three groups. You may write “Group 1 (N=?).

Line 175: instead vaguely mentioning “several physiological variables”, please clearly describe all the physiological variables assessed.

Lines 177-179: “The levels of body awareness, subjective wellbeing and perceived stress level were measured using five validated psychometric tests…”. Why five assessment tools were used to assess three variables mentioned? Also pleas explain why there are seven psychological assessment tools listed under ‘Measures’ and not five? Also the measures described in lines 219-285 do not correspond to the variables mentioned in line 177-179. Eg. FFMQ was administered, but assessment of mindfulness is not given in the variables listed in lines 177-179.

Line 181: did the authors mean “Participants filled….” Also, five questionnaires? Please explain why seven are listed under Measures.

Line 230: Couldn’t find the full form of MET. If you haven’t called it out (mentioning the full form along with acronym at the first usage of an acronym) before, please do so.

The selection procedures for the recruitment of participants to the experimental condition can be represented using a CONSORT flow diagram. Were there any dropouts? And why?

The recruitment process of participants for the qualitative part of the study is not clear. Also please describe more about the need and procedures of qualitative data collection.

Please include a brief description of the administration of both interventions including who have administered both and if the differences in expertise could be a reason for some of the results obtained.

Results:

Lines 356-357: “None of the potential participants who responded to the PARQ+ were excluded from the study”. Why? Also please clearly describe the outcomes of any assessment tools you have used to screen the participants (as exclusion criteria).

Results require substantial changes.

Other than the figures, please include tables containing:

1) Comparison of baseline scores of all three groups on all variables under study. You can use ANOVA, if the data fulfill the parametric test requirements. At least for the variables where it is normally distributed, please report ANOVA results. Also, if the data is not normally distributed, please explain the possible reasons.

2) Comparison of post-test scores of all three groups on all variables under study. Here also, ANOVA should be the first option.

3) T-test result comparing the pre-post results of all variables for all three groups. Eg. Pre-post comparison of mindfulness scores of experimental Group 1.

31 individuals have participated in the quantitative part of the study, but only 22 are there in the qualitative part. Please explain why with the selection procedure.

Discussion:

Line 673 says sample size was 33, but earlier it was given as 31 for quantitative and 22 for qualitative. Please clarify.

Line 688: the meaning of the sub-title is not clear.

Please include the limitations of the study.

Conclusion:

“In our opinion, a combination of both programs could provide a safe framework that would help to increase the levels of teacher wellbeing and give them resources to improve interaction with students, given that in education the quality of interpersonal relationships is an important part of the teaching-learning process”. Both the interventions share many common components as displayed in a table earlier. So what exactly did the authors mean by the application of both program together? Are you proposing a novel intervention? Have you considered the possibility of the outcomes measures of the current study and the outcomes of the interventions being different, thus producing not much significant results?

Another major issue is that, since there already are many highly effective interventions to reduce stress and enhance well-being, what is the purpose of these new bodyfulness interventions? Even the study results do not support much significant impact. So how do the authors justify its contribution to the existing literature?

6. PLOS authors have the option to publish the peer review history of their article (what does this mean?). If published, this will include your full peer review and any attached files.

Reviewer #1: No

Reviewer #2: **Yes: **Allen Joshua George

---

## [Author Response · Author response to Decision Letter 0]

11 Aug 2022

We greatly appreciate the time taken to review the article and the suggestions made. Responses are indicated in this document in red and changes to the manuscript and additional material are provided in accompanying documents.

Reviewer #1: It was a pleasure reading the authors’ work and in their detailed description of their intervention. However, the manuscript requires further re-writing.

First of all, the authors need to provide a shorter and better structured Discussion section. Currently the section is more than 10-pages long and parts of it may be suitable for a dissertation chapter but offer little value to the discussion of the findings of their paper. For example, their opening paragraphs (lines 669-687) in their Discussion focus on their methodology rationale that would be a better bit to their methodology section. Similarly, the first few pages in the Discussion they go on to be a presentation of methodology-related relevant literature that may again be a good fit to a dissertation’s methodology chapter but it offers little to the Discussion of the research findings. My advice to the authors is to reduce significantly the length of that section. They can start with a couple of paragraphs discussing their research objectives and methodology decisions within the scope of relevant literature and the move on to discussing their findings in relation to other studies in teachers or application of these interventions in other populations

We have re-written the discussion section following reviewer’s indications. 

The title of the manuscript mentions that this is a “proposal of intervention”, however their methodology, and results section describe what may be best defined as a “pilot” study. If the authors indeed aimed to report the development of an intervention, then they’d need t restructure the paper to clearly show what parts are relevant to the “development” of the intervention and what elements involve its “delivery”.

We intended to carry out a pilot study, so we have included some modifications to clarify this point in the title, abstract, introduction and methodology (lines 27-31,155, 170-179 and 191-200). Also some modifications have been introduced in methodology and results looking for more clarity.

Reviewer #2: The current study has adopted a mixed method approach – using experimental data and content analysis – to study the efficacy of two intervention on stress, subjective well-being, and self-knowledge of higher education teachers. The authors have wisely opted for mixed method considering the limitation of the study with only quantitative assessment, but effort is needed to combine the outcomes of quantitative and qualitative domains and present it as complimentary to each other. Although the attempt was elaborate with many variables, the study apparently has many limitations that need to be addressed before publication. Especially the variables described at different parts of the manuscript are not congruent. Eg. The study measures listed contain FFMQ, to assess mindfulness, but nowhere in the manuscript the authors have mentioned about mindfulness. Similarly, the variables measured and variables described as the outcomes are not the same. 

We have better explained the variables and the reasons why a mixed approach was chosen (see lines 192-202; lines 246-2502)

Further comments are added here.

Title: Why there are two titles given (1-6)?

The title was adjusted to the requirements of this journal. It is the journal format. 

Abstract: Wordiness can be reduced in the abstract. Eg. in lines 20-22, the word ‘theirs’ is repeated five times, which is unnecessary.

We have rewritten the abstract according the reviewer’s suggestions.

Clearly state the number of groups, ie. two experimental groups and one control group with number of participants in each.

Done

Introduction: In line 57, please include the in-text citation apart from the numbering.

Done (see line 62 now)

Materials and Methods: lines 163-166 – Please explain what PARQ+ did assess. “participants should 166 not have lengthy practice with any of the tools proposed in the program” – was this another exclusion criterion? (Then better to mention exclusion criteria) Does it mean that the participants should not already have lengthy practices? Some more clarity is required here as what the authors have meant, since the intervention is not elaborated in the manuscript so far. Also please define lengthy. What was the duration beyond which a practice was considered lengthy?

We have tried to explain better according reviewer’s suggestion (see lines 170-172). PARQ + is a 7-step self-screening tool that is typically used to determine the safety or possible risks of exercising based on the health history, current symptoms, and risk factors. It is designed to identify the small number of adults for whom physical activity may be inappropriate. If a person answers yes to one or more questions, the individual should complete a thorough follow-up of medical questions and consult a physician about beginning physical activity. 

We consider a lengthy practice any physical activity similar to those implemented in the project practiced during at least six months on a weekly basis. Following the reviewer’s suggestion, we have introduced this as an exclusion criterion (see lines 172-174). 

Lines 171-173: please mention the number of participants in all three groups. You may write “Group 1 (N=?).

Done (now this information is in lines 181-183). 

Line 175: instead vaguely mentioning “several physiological variables”, please clearly describe all the physiological variables assessed.

Done (see now lines 197-199). 

Lines 177-179: “The levels of body awareness, subjective wellbeing and perceived stress level were measured using five validated psychometric tests…”. Why five assessment tools were used to assess three variables mentioned? Also pleas explain why there are seven psychological assessment tools listed under ‘Measures’ and not five? Also the measures described in lines 219-285 do not correspond to the variables mentioned in line 177-179. Eg. FFMQ was administered, but assessment of mindfulness is not given in the variables listed in lines 177-179.

We assessed five psychological variables. Following the reviewer’s suggestion we have detailed these five variables in lines 246-252 (previously lines 177-179).

There are no adapted tools for body movement awareness interventions, reason why authors decided to use five validated psychometric tests, as they covered different aspects on which the programs could have an impact.

Line 181: did the authors mean “Participants filled….” Also, five questionnaires? Please explain why seven are listed under Measures.

Participants filled two questionnaires about physical activity. One of them, the PARQ+ was used as exclusion criteria in the recruitment phase. Five questionnaires were used for psychological variables. See lines 246-252.

Line 230: Couldn’t find the full form of MET. If you haven’t called it out (mentioning the full form along with acronym at the first usage of an acronym) before, please do so.

Done. We had included the full form in Figure 1, but in fact we had not included the first usage of the acronym, so we have resolved this including the full form along with acronym (line 264).

The selection procedures for the recruitment of participants to the experimental condition can be represented using a CONSORT flow diagram. Were there any dropouts? And why?

Following the reviewer’s suggestion, we have included a CONSORT flow diagram as Figure 1 showing the recruitment process. Accordingly, we have renumbered all figures. 

The recruitment process of participants for the qualitative part of the study is not clear. Also please describe more about the need and procedures of qualitative data collection.

There is no different procedure for recruiting participants for the qualitative part. They are the same participants. A slightly change in line 347 tries to clarify this. We have also included this information in the CONSORT flow diagram.

Please include a brief description of the administration of both interventions including who have administered both and if the differences in expertise could be a reason for some of the results obtained.

We have reinforced the explanation requested about the expertise levels of the professionals in charge of the different programs (lines 218-220). We have also included it as a potential limitation (lines 1035-1039).

Results:

Lines 356-357: “None of the potential participants who responded to the PARQ+ were excluded from the study”. Why? Also please clearly describe the outcomes of any assessment tools you have used to screen the participants (as exclusion criteria).

We have added more details in lines 392-396 (previously lines 356-357). The screening tool requires that the participant answers YES to at least one of the 7 initial questions of the test in order to continue with further questions and suggest a medical check-up before doing any kind of physical activity. This was not the case as all participants answered NO to all 7 initial questions.

Results require substantial changes.

Other than the figures, please include tables containing:

1) Comparison of baseline scores of all three groups on all variables under study. You can use ANOVA, if the data fulfill the parametric test requirements. At least for the variables where it is normally distributed, please report ANOVA results. Also, if the data is not normally distributed, please explain the possible reasons.

We have included the results of ANOVA pre-test in the manuscript as Table 5. We have renumbered the rest of the tables accordingly

2) Comparison of post-test scores of all three groups on all variables under study. Here also, ANOVA should be the first option.

We have included the results of ANOVA post-test as a Supplementary file S3.

3) T-test result comparing the pre-post results of all variables for all three groups. Eg. Pre-post comparison of mindfulness scores of experimental Group 1.

We have included the results of T-test in the Supplementary file S3 to complement the figures and text. 

31 individuals have participated in the quantitative part of the study, but only 22 are there in the qualitative part. Please explain why with the selection procedure.

There was a typo mistake in the number. Correct number is 21. Only the participants who participated in the interventions (group 2 and 3, n=11 and n=10, respectively) wrote reflexive diaries (see lines 505-506). We have also added this information in the CONSORT diagram (Figure 1). 

Discussion:

Line 673 says sample size was 33, but earlier it was given as 31 for quantitative and 22 for qualitative. Please clarify.

That was a typo mistake. Total number of participants for the three groups were 31. Only the participants who participated in the interventions (group 2 and 3, n=11 and n=10, respectively) wrote reflexive diaries (see lines 505-506). We have clarified this in the text and through a CONSORT diagram (Figure 1).

Line 688: the meaning of the sub-title is not clear.

The sub-title follows the format requirements stablished by the journal.

Please include the limitations of the study.

We have reinforced the limitations of the study in lines 1015-1042.

Conclusion:

“In our opinion, a combination of both programs could provide a safe framework that would help to increase the levels of teacher wellbeing and give them resources to improve interaction with students, given that in education the quality of interpersonal relationships is an important part of the teaching-learning process”. Both the interventions share many common components as displayed in a table earlier. So what exactly did the authors mean by the application of both program together? Are you proposing a novel intervention? Have you considered the possibility of the outcomes measures of the current study and the outcomes of the interventions being different, thus producing not much significant results?

We have modified the text to try to better explain our point of view. We show differences in two mind-body approaches. We believe that a successful intervention could take advantage of the strengths of both approaches by combining resources from both. In fact, we have carried out another study with engineering students in which we combined the sequential structure of Hatha Yoga with creative movement, play and symbolism in a relational context provided by Dance Movement Therapy (https://doi.org/10.3390/educsci12020111). 

Another major issue is that, since there already are many highly effective interventions to reduce stress and enhance well-being, what is the purpose of these new bodyfulness interventions? Even the study results do not support much significant impact. So how do the authors justify its contribution to the existing literature?

Dance movement therapy is not a new bodyfulness intervention. Its origin is in the 1940 in United States and there are many evidences about its effectivity in mental health. The novelty of this paper lies in its introduction in higher education that until now is scarce. The relational component, not included in others approaches, justify the introduction of this approach in education.

---

## [Decision Letter · Decision Letter 1]

19 Sep 2022

PONE-D-21-31772R1Stress, subjective wellbeing and self-knowledge in higher education teachers: a pilot study through bodyfulness approachesPLOS ONE

Dear Dr. Rodríguez-Jiménez,

Thank you for submitting your manuscript to PLOS ONE. After careful consideration, we feel that it has merit but does not fully meet PLOS ONE’s publication criteria as it currently stands. Therefore, we invite you to submit a revised version of the manuscript that addresses the points raised during the review process.

We look forward to receiving your revised manuscript.

Kind regards,

Maria Armaou, PhD

Guest Editor

PLOS ONE

Journal Requirements:

Additional Editor Comments:

Thank you for submitting a revision of your manuscript. I am the designated Guest Editor to handle this document and I was also part of the first round of the review process. It is clear that the quality of the manuscript has substantially improved and the authors have done a great work addressing the comments raised in the review process. I am satisfied with the authors’ response and revisions to my comments and especially regarding the material that needed to be presented in different sections which allowed a better evaluation of the manuscript.

However, there are still some remaining issues which I believe can be easily amended:

-Currently, the authors do not clearly signpost their outcome measures. Just as the authors explain in their response they used a set of questionnaires to measure specific psychological variables. My recommendation is to rename the title “Measures” (if possible) to “Outcome measures” and the processed to title the scales used as “Quantitative Measures”

- Is it possible to break-down the study objectives? Currently in line 155 only the aim of the pilot study is provided along with a reference in lines 159-162 to the differences between the two approaches.

-The main remaining weakness of the paper is that it offers little insight on the necessity of the qualitative data. Were those data meant to address a specific separate research question? If yes, it would be useful to add it at the end of the introduction section. If, though, they are complementary then the authors would need to elaborate a bit on how they are combined.

-I found very helpful the authors’ link to another piece of their work, within which they mention that they conducted “An experimental pre-post design with a control group was combined with an exploratory qualitative approach”. Does that description also apply to the current study? If yes then I would recommend to rephrase it slightly and use that to describe the current study. Similarly, it would help to separate between “Quantitative measures” and “Qualitative Exploration”.

-In their previous paper that the authors cite in their response to reviewers they mention that “a structured warm up was designed following the characteristics Hatha Yoga Is that an example of combining the two approaches? If yes, then I would recommend adding that argument and possibly pick up on findings from their current study (one example would be enough) on how one could go to combine other elements of those two approaches.

-The authors report that there were some differences in the effects of the two approaches, which was also evident in their qualitative analysis. It would be useful to read what this could possibly mean for the content of future interventions and their targeted outcomes. I believe that they have pointed towards that that in lines 1041-1044: “Furthermore, the choice of assessment instruments seems to be more appropriate for the HY approach. In future work, it would be interesting to incorporate instruments that capture relational aspects as they are essential in the state of life satisfaction, well-being and self-knowledge of the individual”. For this reason, it would be useful to add a sentence in the discussion commenting on the nature of the instruments they used in this study and possibly highlight that the qualitative exploration signals possibly the need to use in future feasibility study other types of outcome measures.

-Finally, I would like to pick up on the authors’ response that “The novelty of this paper lies in its introduction in higher education that until now is scarce. The relational component, not included in others approaches, justify the introduction of this approach in education.”

I believe that this is an important statement that should be added at the end of the introduction. Furthermore, I think it would be greatly beneficial if the authors reflect on other factors that may influence or impact on intervention implementation or effectiveness within higher education. Are there any context factors that they have identified that they would need to address in a future feasibility study? Although their intervention did not adopt a mindfulness-based approach, are there lessons from other types of bodyfulness/mindfulness-based interventions applied in higher education that are also relevant to the implementation and evaluation of the type of intervention that the authors describe? Do they differ to what we know about other interventions aiming to improve wellbeing and reduce stress?

Reviewers' comments:

Reviewer's Responses to Questions

**Comments to the Author**

1. If the authors have adequately addressed your comments raised in a previous round of review and you feel that this manuscript is now acceptable for publication, you may indicate that here to bypass the “Comments to the Author” section, enter your conflict of interest statement in the “Confidential to Editor” section, and submit your "Accept" recommendation.

Reviewer #2: All comments have been addressed

2. Is the manuscript technically sound, and do the data support the conclusions?

Reviewer #2: Yes

3. Has the statistical analysis been performed appropriately and rigorously? 

Reviewer #2: Yes

4. Have the authors made all data underlying the findings in their manuscript fully available?

Reviewer #2: Yes

5. Is the manuscript presented in an intelligible fashion and written in standard English?

Reviewer #2: Yes

6. Review Comments to the Author

Reviewer #2: The authors have addressed all the comments and modified the manuscript. The revised manuscript has better quality and clarity.

7. PLOS authors have the option to publish the peer review history of their article (what does this mean?). If published, this will include your full peer review and any attached files.

Reviewer #2: **Yes: **Allen Joshua George

---

## [Author Response · Author response to Decision Letter 1]

20 Oct 2022

Additional Editor Comments:

Thank you for submitting a revision of your manuscript. I am the designated Guest Editor to handle this document and I was also part of the first round of the review process. It is clear that the quality of the manuscript has substantially improved and the authors have done a great work addressing the comments raised in the review process. I am satisfied with the authors’ response and revisions to my comments and especially regarding the material that needed to be presented in different sections which allowed a better evaluation of the manuscript. However, there are still some remaining issues which I believe can be easily amended

Response: Thank you very much for your comments. We have tried to follow all your indications. 

-Currently, the authors do not clearly signpost their outcome measures. Just as the authors explain in their response they used a set of questionnaires to measure specific psychological variables. My recommendation is to rename the title “Measures” (if possible) to “Outcome measures” and the processed to title the scales used as “Quantitative Measures”

Response: Following the editor’s indications, we have renamed the title “Measures” to “Outcome Measures”. We have left the other titles for the quantitative and qualitative measures as they were in order to respect the limitations of the journal’s guidelines to three heading levels. Anyway, we have introduced additional comments in lines 249 and 256 to clarify the type of measures used. 

- Is it possible to break-down the study objectives? Currently in line 155 only the aim of the pilot study is provided along with a reference in lines 159-162 to the differences between the two approaches.

Response: Following the editor’s suggestions, we have broken-down the general objectives in two specific objectives (see lines 157-161).

-The main remaining weakness of the paper is that it offers little insight on the necessity of the qualitative data. Were those data meant to address a specific separate research question? If yes, it would be useful to add it at the end of the introduction section. If, though, they are complementary then the authors would need to elaborate a bit on how they are combined.

Response: We have tried to better justify the importance of introducing qualitative data at the end of the introduction. In our view, mixed approaches, combining quantitative and qualitative data, enrich and complement the information gathered by shedding more light on processes, and are essential in areas such as the one we are dealing with. Some references have been added to remark this consideration (see lines 162-169).

-I found very helpful the authors’ link to another piece of their work, within which they mention that they conducted “An experimental pre-post design with a control group was combined with an exploratory qualitative approach”. Does that description also apply to the current study? If yes then I would recommend to rephrase it slightly and use that to describe the current study. Similarly, it would help to separate between “Quantitative measures” and “Qualitative Exploration”.

Response: Yes, that description also apply to this study. We think that we have clarified better the design with the additional information added throughout the new version of the paper. 

-In their previous paper that the authors cite in their response to reviewers they mention that “a structured warm up was designed following the characteristics Hatha Yoga Is that an example of combining the two approaches? 

Response: Yes, in the paper cited, some elements from the two approaches were combined. In fact, one was the structured Hatha Yoga warm-up. In the present paper, the two warm-ups were different, each one of them following their own criteria adapted to the respective programs.

If yes, then I would recommend adding that argument and possibly pick up on findings from their current study (one example would be enough) on how one could go to combine other elements of those two approaches.

Response: We have tried to include one specific example in the discussion (see lines 880-886). 

-The authors report that there were some differences in the effects of the two approaches, which was also evident in their qualitative analysis. It would be useful to read what this could possibly mean for the content of future interventions and their targeted outcomes. I believe that they have pointed towards that that in lines 1041-1044: “Furthermore, the choice of assessment instruments seems to be more appropriate for the HY approach. In future work, it would be interesting to incorporate instruments that capture relational aspects as they are essential in the state of life satisfaction, well-being and self-knowledge of the individual”. For this reason, it would be useful to add a sentence in the discussion commenting on the nature of the instruments they used in this study and possibly highlight that the qualitative exploration signals possibly the need to use in future feasibility study other types of outcome measures.

Response: We have added the idea suggested (see lines 923-927) and rename the heading “limitations” to “limitations and future directions” to be coherent with the content.

-Finally, I would like to pick up on the authors’ response that “The novelty of this paper lies in its introduction in higher education that until now is scarce. The relational component, not included in others approaches, justify the introduction of this approach in education.”

I believe that this is an important statement that should be added at the end of the introduction. 

Response: Following the editor’s recommendation, we have included this statement at the end of the introduction (lines 167-169). 

Furthermore, I think it would be greatly beneficial if the authors reflect on other factors that may influence or impact on intervention implementation or effectiveness within higher education. Are there any context factors that they have identified that they would need to address in a future feasibility study? 

Response: You are right. In fact, contextual factors are really important. We have included this consideration in the discussion, lines 910-921.

Although their intervention did not adopt a mindfulness-based approach, are there lessons from other types of bodyfulness/mindfulness-based interventions applied in higher education that are also relevant to the implementation and evaluation of the type of intervention that the authors describe? Do they differ to what we know about other interventions aiming to improve wellbeing and reduce stress?

Response: We have reinforced the importance of relational component to improve wellbeing and reducing stress. We have also underlined the importance of a structured frame (as mindfulness-based interventions offer) that help participants to increase body awareness, body knowledge and self-confidence regarding the own body (see lines 825-827].

---

## [Decision Letter · Decision Letter 2]

16 Nov 2022

Dear Dr.Rodríguez-Jiménez,

Thank you for the submission of your revised manuscript "Stress, subjective wellbeing and self-knowledge in higher education teachers: a pilot study through bodyfulness approaches". After careful review both reviewers agree that the revised manuscript successfully addresses  all the comments earlier raised in the review process and they recommend the manuscript to be accepted for publication.

Kind regards,

Maria Armaou, PhD

Guest Editor

PLOS ONE

Additional Editor Comments (optional):

Thank you for submitting your revised manuscript. I can confirm that the manuscript fully addresses all the issues raised through the review process and is currently fit for publication.

Reviewers' comments:

Reviewer's Responses to Questions

**Comments to the Author**

1. If the authors have adequately addressed your comments raised in a previous round of review and you feel that this manuscript is now acceptable for publication, you may indicate that here to bypass the “Comments to the Author” section, enter your conflict of interest statement in the “Confidential to Editor” section, and submit your "Accept" recommendation.

Reviewer #2: All comments have been addressed

2. Is the manuscript technically sound, and do the data support the conclusions?

Reviewer #2: Yes

3. Has the statistical analysis been performed appropriately and rigorously? 

Reviewer #2: Yes

4. Have the authors made all data underlying the findings in their manuscript fully available?

Reviewer #2: Yes

5. Is the manuscript presented in an intelligible fashion and written in standard English?

Reviewer #2: Yes

6. Review Comments to the Author

Reviewer #2: Since the authors have addressed all the comments from the reviewers during two revisions, which has improved the quality of the manuscript, I suggest acceptance of the manuscript for publication.

7. PLOS authors have the option to publish the peer review history of their article (what does this mean?). If published, this will include your full peer review and any attached files.

Reviewer #2: **Yes: **Allen Joshua George

---

## [Editor Report · Acceptance letter]

21 Nov 2022

PONE-D-21-31772R2 

Stress, subjective wellbeing and self-knowledge in higher education teachers: a pilot study through bodyfulness approaches 

Dear Dr. Rodríguez-Jiménez:

I'm pleased to inform you that your manuscript has been deemed suitable for publication in PLOS ONE. Congratulations! Your manuscript is now with our production department. 

Kind regards, 

on behalf of

Dr. Maria Armaou 

Guest Editor

PLOS ONE